# Hitting the High Notes: Subset Selection for Maximizing Expected Order Statistics

**Aranyak Mehta**
Google Research
aranyak@google.com

**Uri Nadav**
Google Research
urinadav@google.com

**Alexandros Psomas**[*]
Purdue University
apsomas@cs.purdue.edu

**Aviad Rubinstein**
Stanford University
aviad@cs.stanford.edu

## Abstract

We consider the fundamental problem of selecting $k$ out of $n$ random variables in a way that the expected highest or second-highest value is maximized. This question captures several applications where we have uncertainty about the quality of candidates (e.g. auction bids, search results) and have the capacity to explore only a small subset due to an exogenous constraint. For example, consider a second price auction where system constraints (e.g., costly retrieval or model computation) allow the participation of only $k$ out of $n$ bidders, and the goal is to optimize the expected efficiency (highest bid) or expected revenue (second highest bid).

We study the case where we are given an explicit description of each random variable. We give a PTAS for the problem of maximizing the expected highest value. For the second-highest value, we prove a hardness result: assuming the Planted Clique Hypothesis, there is no constant factor approximation algorithm that runs in polynomial time. Surprisingly, under the assumption that each random variable has monotone hazard rate (MHR), a simple score-based algorithm, namely picking the $k$ random variables with the largest $1/\sqrt{k}$ top quantile value, is a constant approximation to the expected highest and second highest value, *simultaneously*.

## 1 Introduction

We study a basic algorithmic meta-question: given $n$ independent random variables, select $k$ of them, with the objective of maximizing the expected largest value, and/or the expected second highest value. We are motivated by the following applications:

**Search engine** Given a search query, the search engine has to return $k$ results of $n$ candidates. The random variables model the uncertainty about the user's utility from each result. Among the $k$, the human can select the most relevant result, and our goal is to maximize their utility. In their seminal WAND paper, Broder et. al. [BCH+03] point out that a search engine's latency constraint prevents it from perfectly scoring all possible candidates, and propose a two-tier solution for scoring documents in a search engine, where first they run a fast approximate evaluation and then a full slower evaluation limited to only promising candidates.

**Procurement auctions** A buyer in a complex procurement auction (e.g. for a large engineering project [RL07, Tan92]) receives $n$ initial proposals. They need to select a subset of $k$ bidders who will be allowed to submit a more detailed second-stage proposal, of which the best will be selected.

---

[*]This work was done while Alexandros Psomas was a visiting scientist at Google research MTV.

**Simple ad auctions** A platform receives $n$ candidates for a slot to display an online ad. The candidates come with a value-per-click bid, as well as a set of features for estimation of relevance and click-through-rate (CTR). A large deep model converts these to CTR estimates ([HPJ+14, MHS+13]), which are combined with the per-click bid to generate an auction score. An auction (typically second price [Var07, EOS07]) is run to choose the ad to display and its per-click payment. In this setting, computational constraints (the auction has to be extremely fast) typically prevent evaluation of the large model on all $n$ candidates; all but $k$ are filtered using scores from a faster, less accurate model, before going to the auction.

**The race for a vaccine** A government agency like NIH or NSF can fund $k$ out of $n$ competing grant proposals that aim to solve the same problem, e.g. develop a vaccine for COVID-19. Ultimately, the best vaccine will be used.

The significance of the expected largest value is clear in all applications. In the context of auctions (of both types), the expected second-largest is important since it is the revenue of a second-price auction. The expected maximum objective was previously studied by Kleinberg and Raghu [KR18] in the context of a fifth application, team selection.

**Team selection** A manager needs to select $k$ out of $n$ applicants to form a team to work on a particular task [KR18]. Every applicant takes one or multiple tests, modeled as samples from the distribution of performance. In the "contest" model, the team's performance is evaluated based on the best outcome of any team member.

[KR18] focused on the existence of good score-based selection rules, i.e. rules that separately compute a score for each variable, and then take the $k$ variables with the highest score. It is tempting, and very common in practice, to compute the average performance for each variable, and then pick the best $k$. But, this would lead to a suboptimal solution. As a simple example, consider a scenario with $n = 20$ candidates: 10 that always score 1.1 (with probability 1) and 10 score 0 with probability 0.9, and 10 otherwise (with probability 0.1). We must form a 10-member team. The first group of candidates has higher individual averages, but the group's expected *maximum* score is only 1.1. On the other hand, the second group has lower individual scores, but the probability that the maximum is less than 10 is $0.9^{10}$; the expected maximum is larger than 6.5. Thus, a group of high variance members can outperform a team formed by the members with the highest individual score [Pag08, HP04].

[KR18] prove that two simple test scores, "best of $k$ samples" and "expectation over $1/k$ top quantile", obtain a constant factor approximation to the expected-maximum objective. Furthermore, they prove that in general, the approximation ratio of any score-based rule is at most a constant (namely $8/9$).

**Our contribution**

In this work, we extend the seminal ideas of [KR18] in multiple directions.

**Algorithms and complexity** We first consider the algorithmic task of computing a near-optimal subset given an explicit description of discrete support random variables, i.e. as a list of (value, probability) pairs. We prove NP-hardness and give a near-linear time PTAS for the highest value objective. Score based algorithms with such performance are ruled out by the lower bound of [KR18], so, of course, our algorithm is not score based. This result shows that looking at the interaction between variables opens the door to much better guarantees. On the other hand, for the second-highest objective, we prove that computing any constant factor approximation is intractable, assuming either (a variant of) the Planted Clique assumption or the Exponential Time Hypothesis.

**A simple and near-optimal score for MHR distributions** In contrast to our worst-case hardness result, we show that if each variable satisfies a *monotone hazard rate (MHR)* assumption,[2] then a simple score-based rule gives a constant factor approximation to both the highest and second-highest objectives, simultaneously. The score $s_i$ of each variable $x_i$ is the value of its $1/\sqrt{k}$ top quantile, namely $s_i = \sup \left\{ \tau : \mathbf{Pr}[X_i > \tau] \geq 1/\sqrt{k} \right\}$.

**Selection rules for machine learning**    In practice, for most of the above scenarios, we don't have an explicit description of the distribution. Relaxing the assumption of access to such an explicit description was left as an open problem in [KR18]. In this paper, we consider a more realistic scenario where each candidate is represented by a vector of features, and the random variables model our uncertainty about the true value of each candidate. We develop regression-based analogs of [KR18]'s and our scoring rules and empirically evaluate them on a neural net to predict the popularity of tweets on Twitter. We observe that the Quantile method and [KR18]'s method have similar performance, and both outperform regression (squared loss), for a large range of input quantiles (including the choices that we have theoretical guarantees for).

**Additional related work by [GGM10, CHL$^+$16, SS20]**

After the publication of the conference version of this paper, we became aware of earlier [GGM10, CHL$^+$16] and concurrent [SS20] works on approximation algorithms for the highest value objective. These works refer to essentially the same problem using the names $k$-**MAX** or **non-adaptive Probe-Max**. Specifically, [CHL$^+$16] give a PTAS for this problem, and [SS20] improve to an EPTAS. We note that our algorithm is also an EPTAS: for a $(1 - \epsilon)$-factor approximation, the running time is

$$O(n \cdot \log^{C(\epsilon)}(k)) \le O(n^{1+o(1)} \cdot 2^{C(\epsilon)^2}),$$

for some $C(\epsilon)$ that depends only on $\epsilon$. On the complexity side, the NP-hardness for exact algorithms for the highest value objective follows from [GGM10, CHL$^+$16].

## 2    Model

There is a set $\mathcal{N} = \{X_1, \ldots, X_n\}$ of $n$ mutually independent random variables. We write $[n]$ for the set $\{1, \ldots, n\}$. Our goal is to select a subset $S \subset [n]$ of size $k \ge 2$ in order to maximize the expected largest value, denoted by $\mathsf{E}[\max_{i \in S} X_i]$, and expected second largest value, denoted by $\mathsf{E}[\mathrm{smax}_{i \in S} X_i]$. Let $\mathrm{OPT}_{max}(X) = \max_{S \subset [n]:|S|=k} \mathsf{E}[\max_{i \in S} X_i]$ and $\mathrm{OPT}_{smax}(X) = \max_{S \subset [n]:|S|=k} \mathsf{E}[\mathrm{smax}_{i \in S} X_i]$ be the optimal expected largest and second largest values. We often overload notation and refer to the optimal subsets themselves as $\mathrm{OPT}_{max}(X)$ and $\mathrm{OPT}_{smax}(X)$. Also, when clear from context we drop the subscript, and simply write $\mathrm{OPT}(X)$.

In Sections 3 and 4 we are interested in computation: given an explicit description of the $X_i$s, i.e. for each $X_i$ pairs of numbers $(v_j, p_j)$ indicating the probability $p_j$ that $X_i$ takes value $v_j$, can we compute a good approximation to $\mathrm{OPT}_{max}(X)$ and $\mathrm{OPT}_{smax}(X)$? In Section 5 we consider a slightly different model, where each $X_i$ is a continuous random variable. Let $F_i(x)$ and $f_i(x)$ be the cumulative distribution function (CDF) and probability density function (PDF) of $X_i$. We will be interested in a special family of random variables.

**Definition 1** (MHR)**.** *A random variable $X$ has Monotone Hazard Rate (MHR) if its hazard rate $h(v) = \frac{f(v)}{1-F(v)}$ is a monotone non-decreasing function.*

Many common families of distributions such as the Uniform, Exponential, and Normal have monotone hazard rate. MHR distributions have been extensively studied in the statistics literature under the (perhaps better) name of IFR, Increasing Failure Rate (see [BP96]) but to maintain consistency with the computer science literature we refer to them as MHR in this paper.

## 3    A PTAS for Expected Largest Value

In this section we study the problem of maximizing the expected largest value. First, we show that the problem is NP-hard.

**Theorem 1.** *Given $n$ random variables $X_1, \ldots, X_n$, an integer $k$ and a target $C$, deciding if there exists a subset of random variables, of size $k$, whose expected largest largest value is at least $C$, is an NP-hard problem.*

We defer the proof to Appendix A. Our main result for this section is a PTAS for maximizing the expected largest value.

**Theorem 2.** *For every fixed $\epsilon \in (0, 1]$ there exists an algorithm that runs in time polynomial in $n$ and $k$, and outputs a $(1 - \epsilon)$ approximate solution to the expected maximum objective.*

Our algorithm uses a number of non-trivial pre-processing steps to simplify every random variable $X_i$ to a new random variable $T_i$ that can be completely described via one of constantly many vectors (this constant, of course, depends on $\epsilon$). After this transformation, the search space is small enough for a brute-force approach to work, by trying all ways to put $k$ "balls", the random variables, into a constant number of "bins", the different descriptions, resulting in a polynomial time algorithm. We can further reduce this to an almost linear time algorithm. We briefly sketch the main ideas. Missing proofs can be found in Appendix B.

Our pre-processing works as follows. First, for some appropriately chosen threshold $\tau$, we replace, for each random variable $X_i$, the outcomes (i.e. points of the support) of $X_i$ with value greater than $\tau$ with a point mass of the same expectation. That is, we construct a new random variable $\hat{X}_i$ that is equal to $X_i$ when $X_i \leq \tau$ and otherwise randomizes between zero and a value $H_{max}$ (formally defined in the appendix), in a way that $\mathsf{E}[\hat{X}_i|\hat{X}_i > \tau] = \mathsf{E}[X_i|X_i > \tau]$. We show (Claim 2) that for any subset of variables, this transformation has a negligible effect on the expected maximum value.

Second, for each random variable $X_i$, we discard outcomes with value smaller than $\epsilon^2\tau$. Those values have a negligible contribution to the expected largest value anyway (Claim 3). Third, the new random variables are supported in the range $[\epsilon^2\tau, \tau] \cup \{0, H_{max}\}$ for each $X_i$. We partition the $[\epsilon^2\tau, \tau]$ range into $\ell := \log_{1-\epsilon}(\epsilon^2) = O(\frac{1}{\epsilon}\log(\frac{1}{\epsilon})) = \tilde{O}(1/\epsilon)$. Let $I_j = [\frac{\epsilon^2}{(1-\epsilon)^{j-1}}\tau, \frac{\epsilon^2}{(1-\epsilon)^j}\tau)$. We further round down the values within each interval $I_j$ to its lower endpoint $\frac{\epsilon^2}{(1-\epsilon)^{j-1}}\tau$, losing a $1-\epsilon$ factor. Thus far we have constructed random variables that are $\ell + 2$ point masses (with the last two corresponding to 0 and $H_{max}$ from Step 1). Fourth, for some appropriately chosen threshold $\eta$, we decompose each variable into a *core* random variable $C_i$ and a *tail* random variable $T_i$ using $\eta$ as the cutoff. We show (Lemma 3) that we can set aside a small portion of our "budget" $k$ to cover almost the full contribution to the expected maximum from the cores using a simple greedy algorithm. We can therefore focus on optimizing the tail random variables. Fifth, for each tail random variable $T_i$, we discard all intervals whose marginal contribution is much smaller than the total expectation from $T_i$. We show (Claim 4) that this step has a negligible effect on the expected maximum of any subset. For each of the remaining intervals, we consider its marginal contributions relative to the total expectation, and round it to the nearest power of $(1 + \epsilon)$.

This concludes the pre-processing. After the last step, we use a new representation for each tail random variable $T_i$, as follows. Let $\mathcal{I} = \cup_{j=1}^{\ell} I_j \cup H_{max}$ be the set of intervals $T_i$ can take a value in. We can write the expectation of $T_i$ as $\mathsf{E}[T_i] = \sum_{I \in \mathcal{I}} \mathbf{Pr}[T_i \in I]\mathsf{E}[T_i|T_i \in I]$. We henceforth use $\mathrm{REL}(i)$ to denote the vector of length $\ell + 1$, whose $j$-th component is the relative contribution of $I$, the $j$-th interval in $\mathcal{I}$, to the expectation of $T_i$. We overload notation and use $I$ for the index of interval $I$. Thus, we have $\mathrm{REL}(i, I) := \frac{\mathbf{Pr}[T_i \in I]\mathsf{E}[T_i|T_i \in I]}{\mathsf{E}[T_i]}$. Notice that $T_i$ is completely described by $\mathsf{E}[T_i]$ and $\mathrm{REL}(i)$. Given our last pre-processing step $\mathrm{REL}(i, I)$ only take a constant number of values. The length of $\mathrm{REL}(i)$ is $\ell + 1 = \tilde{O}(1/\epsilon)$, again, a constant; therefore the total number of $\mathrm{REL}(i)$ vectors is a constant $C(\epsilon)$. Given two random variables with the same $\mathrm{REL}(i)$ vector, it is always preferable to pick the one with the larger expectation (since it stochastically dominates).

Thinking of each different $\mathrm{REL}(i)$ vector as a type, each random variable has one of $C(\epsilon)$ types. At this point, we can simply try all ways to put $k$ "balls", the random variables, into $C(\epsilon)$ "bins", the different types, and taking the best one (of the ones corresponding to feasible assignments with respect to the random variables we actually have). This gives a $O(n)k^{C(\epsilon)}$ time algorithm (where the $O(n)$ comes from the running time of the pre-processing steps). We show how to vastly improve the running time by considering only $\log(k)$ possibilities for each type. Specifically, instead of considering putting $1, 2, \ldots, k$ "balls" to bin $I$, we consider $1, (1 + \epsilon), (1 + \epsilon)^2, \ldots, k$ "balls". The running time is improved to $O(n(\log k)^{O(C(\epsilon))}) = O(n\mathsf{polylog}(k))$.

## 4    Hardness for Expected Second Largest Value

In this section we prove that, in stark contrast to expected maximum, maximizing the expected second largest value is hard to approximate, assuming the planted clique hypothesis or the exponential time hypothesis. The planted clique hypothesis states that there is no polynomial time algorithm that can distinguish between an Erdős-Rényi random graph $G(n, 1/2)$ and one in which a clique of size

polynomial in $n$ (e.g. $n^{1/3}$) is planted. The exponential time hypothesis (ETH) states that no $2^{o(m)}$ time algorithm can decide whether any $3SAT$ formula with $m$ clauses is satisfiable.

**Theorem 3.** *Assuming the exponential time hypothesis or the planted clique hypothesis, there is no polynomial time algorithm that, given $n$ random variables $X_1, \ldots, X_n$, finds a subset of size $k$ whose expected second largest value is a constant factor of the optimal.*

We give a reduction from the densest $\kappa$-subgraph problem, which is known to be hard under both hypotheses [Man17, AAM$^+$11]. We briefly sketch the construction and intuition here, and defer the details to Appendix C.

Given a graph $G$ on $n$ vertices we construct $n$ random variables $X_1, \ldots, X_n$. For every edge $e = (i, j)$ in the graph, we add the value $p_e^2$ with probability $1/p_e$ to the support of $X_i$ and $X_j$, for some value $p_e$. If both $X_i$ and $X_j$ are in a subset, and an edge $(i, j)$ exists, then the second largest value is (exactly equal to) $p_e^2$ with probability at least $1/p_e^2$, which contributes 1 to the expected second largest value. Furthermore, by picking the $p_e$s very far apart, we can ensure the probability that the second largest value is $p_e^2$ but the largest value is strictly larger is negligible. Therefore, the overall expected second largest value for a subset $S$ is roughly the corresponding number of edges in the graph.

# 5 Quantile Based Algorithm

In this section, we consider continuous random variables that have monotone hazard rate. Omitted proofs can be found in Appendix D. Let $\alpha_p^{(i)} \geq \inf\{x | F_i(x) = 1 - \frac{1}{p}\}$, for $p \geq 1$.

**Theorem 4.** *Picking the $k$ random variables with the highest $\alpha_p^{(i)}$, for $p = \sqrt{k}$, is a $32$ approximation to the optimal subset for the expected largest value and a $1000$ approximation to the optimal subset for the expected second largest value.*

We note that we did not try to optimize the constant factors, and further improvements could be possible. Let $S$ be the subset selected by the algorithm. Let $\hat{X}_i$ be the random variable that is identical to $X_i$ up until $\alpha_{\sqrt{k}}^{(i)}$, and takes value $\alpha_{\sqrt{k}}^{(i)}$ with probability $\frac{1}{\sqrt{k}}$. We analyze the algorithm in two steps.

First, in Section 5.1 we show that $S$ is an almost optimal subset for the truncated random variables (Lemma 1), i.e. for some small $\delta_k, \delta_k' > 0$, $\mathsf{E}[\max_{i \in S} \hat{X}_i] \geq (1 - \delta_k)\mathsf{E}[\max_{i \in A} \hat{X}_i]$ and $\mathsf{E}[\mathrm{smax}_{i \in S} \hat{X}_i] \geq (1 - \delta_k')\mathsf{E}[\mathrm{smax}_{i \in B} \hat{X}_i]$, for all $A, B \subseteq [n]$, $|A| = |B| = k$. Second, in Section 5.2 we show that by truncating at $\sqrt{k}$ we only lose constant factors (Lemma 2). Given the two lemmas, we complete the proof of Theorem 4 in Section 5.3.

## 5.1 Almost optimal selection for truncated random variables

We start by showing that for truncated random variables we can make an almost optimal selection. The intuition is as follows. Let $\alpha_{min}$ be the $k$-th largest $\alpha_p^{(i)}$ value. For the random variables in $S$, the probability that each of them exceeds $\alpha_{min}$ is at least $1/p$. In fact, the largest one exceeds $S$ with probability at least $1 - \prod_{i=1}^{k}(1 - 1/p)$. If $p \in \Omega(1/\sqrt{k})$, then with high probability both the largest and the second largest value exceed $\alpha_{min}$. Conditioned on this event, the set $S$ we have chosen contains the random variable with the highest value and the random variable with the second highest value among all $(n)$ random variables.

**Lemma 1.** *Let $\hat{X}_i$ be the random variable that takes value $x$ when (the possibly non MHR) random variable $X_i$ takes value $x$, for all $x < \alpha_p^{(i)}$, and takes value $\alpha_p^{(i)}$ when $X_i$ takes value at least $\alpha_p^{(i)}$ (i.e. with probability $1/p$). Let $S$ be the subset of random variables, $|S| = k$, with the largest $\alpha_p$ values. Then for all $A \subseteq [n]$, $|A| = k$, (1) $\mathsf{E}[\max_{i \in S} \hat{X}_i] \geq \left(1 - (1 - 1/p)^k\right) \mathsf{E}[\max_{i \in A} \hat{X}_i]$, and (2) $\mathsf{E}[\mathrm{smax}_{i \in S} \hat{X}_i] \geq \left(1 - (k+1)(1 - 1/p)^{k-1}\right) \mathsf{E}[\mathrm{smax}_{i \in A} \hat{X}_i]$.*

## 5.2 Loss from truncation

In this section we bound the ratio between the expected highest and expected second highest value between $X_i$ and $\hat{X}_i$, for any subset $A$ of size $k$. For ease of notation we, without loss of generality,

consider the subset $A = [k]$. We consider an algorithm that, given as inputs $k$ random variables $X_1, \ldots, X_k$ outputs anchoring points $\beta_1$ and $\beta_2$ (Algorithm 1). If the variables are MHR then the contribution to $\mathsf{E}[\max_i X_i]$ and $\mathsf{E}[\mathrm{smax}_i X_i]$ from the tail, formally events larger $\beta = \max\{\beta_1, \beta_2\}$ and $\beta_1$, respectively, is upper bounded by (roughly) a constant times $\beta$ and $\beta_1$, respectively. Second, the outputs of this algorithm satisfy, even for non-MHR random variables, that the probability of $\max_i X_i$ and $\mathrm{smax}_i X_i$ being above $\beta$ and $\beta_1$ is at least a constant. Finally, the outputs $\beta_1$ and $\beta_2$ when the algorithm is executed on input $X_1, \ldots, X_k$ and on input $\hat{X}_1, \ldots, \hat{X}_k$ (as defined above, i.e. $\hat{X}_i$ is $X_i$ truncated at $\alpha^{(i)}_{\sqrt{k}}$) are exactly the same. The upper bound on the tail connects $\beta_1, \beta_2$ with the expectations of the original random variables, while the lower bound on the probability (plus Markov's inequality) connects $\beta_1$ and $\beta_2$ with the expectations of the truncated random variables. Combining all these ingredients we get the main lemma for this step.

**Lemma 2.** *Let $X_1, \ldots, X_k$ be MHR random variables. Let $\hat{X}_i$ be the random variable that is identical to $X_i$ up until $\alpha^{(i)}_{\sqrt{k}}$, and takes value $\alpha^{(i)}_{\sqrt{k}}$ with probability $1/\sqrt{k}$. Then $\mathsf{E}[\max_i X_i] \leq 28.8 \mathsf{E}[\max_i \hat{X}_i]$ and $\mathsf{E}[\mathrm{smax}_i X_i] \leq 122 \mathsf{E}[\mathrm{smax}_i \hat{X}_i]$.*

Algorithm 1 is a modification of an algorithm of Cai and Daskalakis [CD15]. Verbatim, their result states that for $k$ independent MHR random variables, $X_1, \ldots, X_k$, there exists an algorithm that outputs an anchoring point $\beta$ such that $\mathbf{Pr}[\max_i X_i \geq \beta/2] \geq 1 - \frac{1}{\sqrt{e}}$ and $\int_{2\beta \log_2(1/\epsilon)}^{\infty} x f_{max_i X_i}(x) dx \leq 36\beta\epsilon \log_2(1/\epsilon), \forall \epsilon \in (0, 1/4)$, where $f_{max_i X_i}(x)$ is the probability density function of $\max_i X_i$. For our purposes, this high level view is not sufficient. This theorem gives us a value $\beta$ such that truncating the $X_i$s at $2\beta \log(1/\epsilon)$ has a small effect on the expected maximum of the $X_i$s. This fact is very surprising, but on first glance seems of little use here. First, we do not know which subset of $[n]$ to use to compute $\beta$ (selecting a good subset is, in fact, the problem we're trying to solve). Second, it is unclear how to use this information to bound $\mathsf{E}[\max_i X_i]/\mathsf{E}[\max_i \hat{X}_i]$. Third, this theorem tells us nothing about the expected second largest value. We need a more flexible approach.

Taking a closer look at their proof, the algorithm of [CD15] looks at quantiles of the form $2^t/k$. Specifically, in round $t$, for $t = 0, \ldots, \log_2 k - 1$, it sorts the remaining random variables by $\alpha_{k/2^t}$ and eliminates the bottom half, keeping track of $\beta_t$, the smallest threshold among surviving random variables. $\beta$ is the maximum of the $\beta_t$s and the $\alpha_2$ value of the unique surviving random variable. Truncating our random variables at $\alpha_k$ and then executing this algorithm for $X_i$ and $\hat{X}_i$ would give the same $\beta$. Unfortunately, such a truncation point is not good enough for the bound on the expected second largest value in Lemma 1. Our first modification is to instead focus on the $\sqrt{2^t/k}$ top quantile values. This guarantees that the algorithm does not use any information from the parts where $X_i$ and $\hat{X}_i$ differ. In order to take care of both $\max_i X_i$ and $\mathrm{smax}_i X_i$ at the same time, further modifications in the book-keeping (which values to remember at each round) and the analysis are necessary.

---

**Algorithm 1:** Algorithm for finding $\beta$

---

**Input:** $\alpha^{(i)}_q$ for $i = 1, \ldots, k$, and $q = \sqrt{k}/\sqrt{2}^t$, for $t = 0, \ldots, \log_2 k - 1$.
Define the permutation $\pi_0(i) = i$, $i \in [k]$. Let $Q_0 = [k]$.
**for** $t = 0, \ldots, \log_2 k - 1$ **do**

> For $j \in [k/2^t]$, sort the numbers $\alpha^{(\pi_t(j))}_{\sqrt{k}/\sqrt{2}^t}$ in decreasing order $\pi_{t+1}$ such that
> $\alpha^{(\pi_{t+1}(1))}_{\sqrt{k}/\sqrt{2}^t} \geq \alpha^{(\pi_{t+1}(2))}_{\sqrt{k}/\sqrt{2}^t} \geq \cdots \geq \alpha^{(\pi_{t+1}(k/2^t))}_{\sqrt{k}/\sqrt{2}^t}$ ;
> $Q_{t+1} = \{\pi_{t+1}(i) | i \leq k/2^{t+1}\}$ ;
> $\beta_t = \alpha^{(\pi_{t+1}(k/2^{t+1}+1))}_{\sqrt{k}/\sqrt{2}^t}$ ;

**end**

Set $\beta_{\log_2 k} = \alpha^{(\pi_{\log_2 k}(1))}_{\sqrt{2}}$ ;
Output $\beta_1 = \max_{t=0,\ldots,\log_2 k - 1} \beta_t$ and $\beta_2 = \beta_{\log_2 k}$;

---

Overall our algorithm works as follows. In round $t$, for $t = 0, \ldots, \log_2 k - 1$, it sorts the random variables by threshold $\alpha_{\sqrt{k/2^t}}$ and eliminates the bottom half. We record the largest threshold among the eliminated random variables. The maximum of these records is $\beta_1$, the threshold we use for the

second highest value $\mathrm{smax}_i\, X_i$. $\beta_2$ is the threshold $\alpha_{\sqrt{2}}$ for the unique random variable that survived the $\log_2 k - 1$ rounds of elimination. The maximum of $\beta_1$ and $\beta_2$ is the threshold we use for the highest value $\max_i X_i$. We assume without loss of generality that $k$ is a power of 2; we can always add random variables that take value deterministically zero.

We bound the contribution to the tail above $\beta_1$ and $\beta_2$ separately in Appendix D.2.1. In Section D.2.2 we lower bound the probability that the maximum and second maximum is above $\max\{\beta_1, \beta_2\}$ and $\beta_1$, respectively; importantly these lower bounds hold even if the random variables are not MHR. We complete the proof of Lemma 2 in Appendix D.2.3.

## 5.3 Putting everything together

*Proof of Theorem 4.* Let $S$ be the subset of $[n]$ selected by our algorithm, i.e. the set of random variables with the largest $\alpha_{\sqrt{k}}$. Let $W^*$ be the subset of $[n]$ that maximizes the expected maximum and $R^*$ be the subset that maximizes the expected second maximum.

$$\mathsf{E}[\max_{i \in S} \hat{X}_i] \geq^{(Lem.\ 1)} \left(1 - (1 - 1/\sqrt{k})^k\right) \mathsf{E}[\max_{i \in W^*} \hat{X}_i]$$

$$\geq 0.91 \mathsf{E}[\max_{i \in W^*} \hat{X}_i] \geq^{(Lem.\ 2)} \frac{0.91}{28.8} \mathsf{E}[\max_{i \in W^*} X_i],$$

where in the second inequality we lower bounded for the value at $k = 2$. Similarly,

$$\mathsf{E}[\mathrm{smax}_{i \in S}\, \hat{X}_i] \geq^{(Lemma\ 1)} \left(1 - (k+1)(1 - 1/\sqrt{k})^{k-1}\right) \mathsf{E}[\mathrm{smax}_{i \in R^*}\, \hat{X}_i]$$

$$\geq 0.122 \mathsf{E}[\mathrm{smax}_{i \in R^*}\, \hat{X}_i] \geq^{(Lemma\ 2)} \frac{0.122}{122} \mathsf{E}[\mathrm{smax}_{i \in R^*}\, X_i] = \frac{1}{1000} \mathsf{E}[\mathrm{smax}_{i \in R^*}\, X_i]. \quad \square$$

# 6 Experiments

We run two types of experiments to evaluate the methods described above; we restrict attention to the simple score-based methods and exclude the more complex PTAS from Section 3. First, we evaluate the methods on synthetic data. That is, we construct explicit distributions that we give as inputs to our methods and measure the expected largest and expected second largest value. We observe that the vast differences in approximation factors do not appear. In other words, despite the poor approximation guarantees of the quantile method in theory, in practice it does just as well as the theoretically superior (better approximation guarantee without the MHR assumption, at least for expected maximum) method of [KR18]. In the same type of experiment, we slightly deviate from measuring the expected highest and second highest value, and compare the methods in a different dimension: how the scarcity of data affects each method's selection. Second, we evaluate the methods on real data, and specifically likes-data from Twitter (from [Kag20]). In practice explicit distributions typically only arise if we fit a model to data. A slightly more realistic assumption is historical samples from the same distribution. In our experiments, we go one step further: we consider the practical scenario where we observe only one value for each feature vector. Here, we have an implicit distribution over our uncertainty. We develop regression-based analogs of the score-based algorithms and compare them. We include some additional figures and details about the implementations in Appendix E.

**Synthetic data.** We construct $n = 500$ (independent but non-identical) Normal distributions $\mathcal{N}_i(\mu_i, \sigma_i)$, where each mean $\mu_i$ is drawn from $U[0, 60]$ and $\sigma_i$ is drawn from $U[0, 30]$. Since we want to deal with non-negative and bounded support, we further *clip* the distributions as follows: for each $\mathcal{N}_i$, we make 5000 draws, taking a min with $V_{max} := 1000$ and a max with 0, and then take the empirical distribution. We note that this process yields an *explicit* distribution that we can give as input to each method. We run this process 100 independent times and compare the following methods, for three different values of $k = 10, 20, 30$: (1) Quantile, the algorithm from Section 5, (2) KR, the algorithm from [KR18], (3) Expectation: pick the $k$ distributions with the highest expected values, and (4) Greedy Submodular Optimization: Pick distributions iteratively, picking the next distribution to maximize the increment in expected reward. This is the standard greedy $(1 - 1/e)$-approximation algorithm from submodular optimization. It is relevant here since the expected maximum objective is a submodular function (see Appendix B or [KR18] for a proof). The Quantile and KR algorithms are

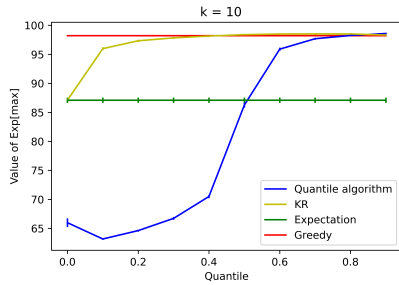

Figure 1: Comparing the average performance (errors bars show standard deviation divided by square root of number of experiments) of the score-based algorithms and Greedy, for selecting $k$ out of $n = 500$ distributions, for the expected maximum objective.

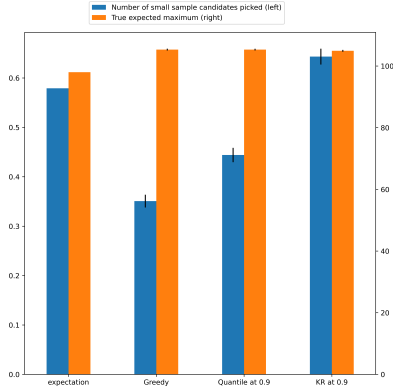

Figure 2: Percentage of small data candidates selected and expected maximum for different algorithms.

parameterized by the quantiles picked. Note that "quantile" is used to refer to the bottom quantile. So, for example, the correct instantiation of the KR method would be to use the $1 - 1/k$, the value such that a $1 - 1/k$ fraction of entries is below. We choose a range of quantiles for each method, and observe the performance under each one.

The results for the expected maximum objective are presented in Figure 1. We include figures for the expected second largest value objective in Appendix E. We can see that the KR algorithm is always outperforming expectation, while the Quantile algorithm's performance is more sensitive to the quantile selected. However, despite the poorer worst-case approximation guarantees of the Quantile algorithm, it performs just as well as the algorithms with better guarantees. For the parameter choices that we have theoretical guarantees for, though, ($1 - \frac{1}{\sqrt{k}}$ for Quantile and $1 - 1/k$ for KR) the two algorithms, as well as the greedy algorithm, are indistinguishable in terms of performance.

**More versus fewer data.** We also run the following "selection-bias" experiment on synthetic data. In the experiments so far, we drew samples from a Normal distribution $\mathcal{N}_i(\mu_i, \sigma_i)$, took the empirical distribution, and used that as the input to our algorithms. The expected largest/second largest value is one measure that we can use to compare the different methods. In theory, improving the objective function is always a better outcome. In practice, in particular in the context of the broader impact of machine learning research, it is important to explore the bias introduced by different algorithms. Algorithmic bias due to *data scarcity* is a well-documented bias in practical ML (e.g. [MMS+19]). Here, we explore the bias of each method with respect to the number of samples available from each distribution. After sampling $\mu_i$ and $\sigma_i$ for each Normal, we also sample a binary label $\{l, m\}$, with probability $1/2$. If the label is $m$ our algorithms see $5000$ samples from this random variable, as before. If the label is $l$ they only see $10$. We compare each method along two metrics: in terms of the percentage of small labeled distributions selected, and in terms of the true expected maximum of the subset selected. We notice that all methods have comparable performance in terms of expected maximum, but select very different candidates in terms of their labels. See Figure 2, and additional figures in Appendix E.

**Real data and the regression-based algorithms.** In most practical situations, we do not have access to an explicit distribution. Instead, we have multi-dimensional feature vectors associated with each data point. To apply the insights from our algorithms to this kind of data, we develop regression-based analogs of the score-based methods (Quantile and KR), and evaluate them together with the standard squared loss algorithm (which naturally corresponds to picking the $k$ candidates with the $k$ largest expected values).

We start with a dataset of $8$ million tweets, sorted in chronological order. We use the first $2$ million for collecting features: we drop all entries with fewer than $5$ likes and pre-process the text, and use as

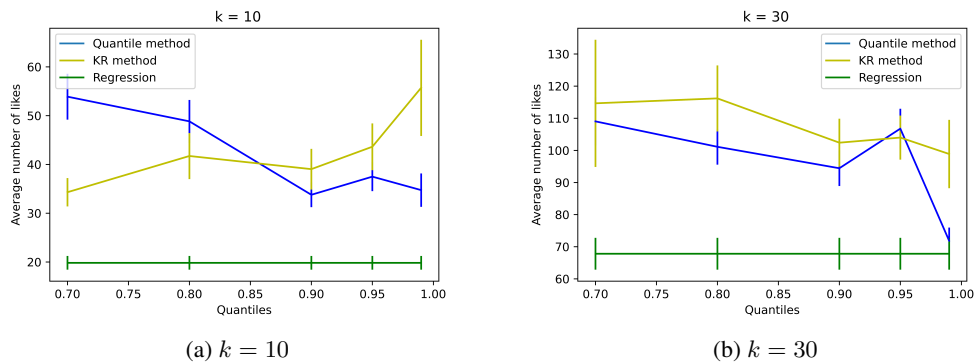

(a) $k = 10$                                        (b) $k = 30$

Figure 3: Comparing the average performance (errors bars show standard deviation divided by square root of number of experiments) of the KR, quantile and regression methods, for selecting $k$ out of $n = 500$ tweets, for the expected maximum objective.

features the (distinct) words that appear in a certain range. This step gives us $\approx 5500$ features. The value in our case is the number of likes a tweet received.

Given a feature vector, there is some correct distribution over the value. At run time, we would like to have an explicit description of these distributions that we can use as inputs to our algorithms, and select a good subset of tweets. Despite the lack of such explicit descriptions, notice that our methods do not use full access to the explicit description. The Quantile method only needs access to a specific quantile, the method that picks the $k$ candidates with the largest expected value only needs the expected value, while the KR method only needs the expected value above a certain quantile. Here we replace exact access (or even sample access) of this information, and instead work with estimates produced by learning algorithms (that are trained using feature vector, number of likes pairs). For the Quantile method we need to estimate a specific quantile, which becomes the usual quantile loss. For the "Regression" method we train using squared loss. The KR-based regression works as follows. The ideal score for the KR algorithm is of the form $\mathsf{E}[X_i | X_i$ in top $q$ quantile$]$. Our implementation first trains using quantile loss on a number of different quantiles. We filter the data using these quantile models, throwing away all entries with real value ("likes") below the prediction. For the remaining entries we train using squared loss.

We train our models using the next 2 million tweets (without dropping any entries). We train a neural network, with 2 hidden layers, with quantile loss at quantiles $[0.7, 0.8, 0.9, 0.95, 0.99]$ (and this is the estimator utilized by the Quantile method). We also train a similar network with squared loss (this is the estimator for the Regression method). For the KR method we first filter the data using the quantile models, and then use a neural net with squared loss for the rest. Lastly, we use the remaining 4 million tweets, that we map to our features, for testing. We randomly perturb this data, and split in non-overlapping chunks of $n = 500$ tweets each. A single experiment samples a chunk of entries, ranks by each method's score and then reveals the true largest/second largest number of likes in the entries picked by each method. We do 8000 experiments.[3] The results are presented in Figure 3. We can see that both the Quantile method and the KR method outperform regression (note that the parameters for which we have theoretical guarantees are $q = 1 - 1/\sqrt{10} \approx 0.7$ for quantile and $q = 1 - 1/10 = 0.9$ for KR).

## Broader impact

Our results apply to a wide range of resource allocation problems in society. The applications stated in the introduction — search, auctions, vaccine development, or team selection — all have broad benefits.

In this work we focused on optimizing the expected maximum subject to the uncertainty about outcome of each random variable. An interesting question arising from our work is how the availability

of more data affects the choices of the algorithms we consider. This question has broad implications given recent research that shows that machine learning training sets are often biased and include less samples corresponding to underrepresented subgroups of the population (e.g. [MMS$^+$19]). Sparseness of data may have two competing effects on the probability of a candidate to be selected: On one hand, our algorithms favor high variance variables, so candidates with less data may be more likely to be selected. On the other hand, our algorithms focus on values in the far tail, and candidates with sparse data may not have any values high enough to show their potential. It is an interesting question to understand how these two affects balance.

As an initial step toward understanding how availability of data affects the probability of a candidate being selected, we consider the following experiment: We take $n = 500$ normal random variables with parameters drawn independently from the same distribution as in Section 6. We randomly partition the variables into *More-Data* and *Less-Data* types. For the More-Data types, we take the empirical distribution from 5000 samples; for Less-Data types, we use 10 samples. We then run different algorithms for selecting $k = 30$ variables using the empirical distribution. For each algorithm, we measure the fraction of Less-Data types selected among the $k$ winners, and the overall max when drawing a fresh sample from each of the $k$ true distributions. We observe (Figure 2) that in terms of expected max (our main objective function), Greedy, Quantile-0.9, and KR-0.9 all perform almost equally well, and Expectation is close after. But Greedy and to some extent Quantile-0.9 under-select Less-Data types, whereas KR-0.9 selects Less-Data types at a higher rate than their fraction (50%) of the population.

## Acknowledgments and Disclosure of Funding

Aviad Rubinstein is supported by NSF CCF-1954927.

## Footnotes

[2]A random variable is MHR if its hazard rate $h(x) = f(x)/(1 - F(x))$ is monotone non-decreasing; see Section 2. Many common families of distributions are MHR, e.g. Normal, Exponential and Uniform.

[3]We removed one outlier tweet with $280,000$ likes (incidentally, Quantile at $0.99$ did include this tweet in its set, even for $k = 10$). The second largest number of likes was $40,000$.

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
