[Supplementary Material]

# A  Proof of Theorem 1

*Proof.* We reduce from INDEPENDENT SET for regular (i.e. equal degrees) graphs to our problem. Let $G = (V, E)$ be an undirected, regular graph on $n$ vertices and $m$ edges.

For every vertex $v \in V$ we construct a random variable $X_v$. The support of $X_v$ includes the value $m^{4p_{u,v}}$, if the graph contains the edge $(u, v)$, and this value occurs with probability $m^{-2p_{u,v}}$, where $p_{u,v}$ is an integer specific to the edge $(u, v)$. Concretely, we can arbitrarily number the edges $e_1, e_2, \ldots$ and set $p_{u,v} = i$ when $e_i = (u, v)$. Notice that $\mathbf{Pr}[X_v = p_{u,v}]^2 \cdot p_{u,v} = 1$. We normalize, by adding a balancing term. Each random variable $X_v$ has some probability $(O(m^{-8m}))$ of taking value $m^{10m}$, in a way that the expectation of every random variable is the same number $\mu$.

**Completeness**  Let $S$ be an independent set of size $k$ in $G$. Then,

$$\mathsf{E}[\max_{v \in S} X_v] = \sum_{v \in S} \sum_{e=(u,v) \in E} m^{4p_{u,v}} \cdot \mathbf{Pr}[X_v = m^{4p_{u,v}}] \cdot \mathbf{Pr}[\forall z \in S, z \neq v : X_z < m^{4p_{u,v}}]$$

$$= \sum_{v \in S} \sum_{e=(u,v) \in E} m^{2p_{u,v}} \cdot (1 - \mathbf{Pr}[\exists z \in S, z \neq v : X_z \geq m^{4p_{u,v}}])$$

$$\geq \sum_{v \in S} \sum_{e=(u,v) \in E} m^{2p_{u,v}} \cdot (1 - \sum_{z \in S: z \neq v} \mathbf{Pr}[X_z \geq m^{4p_{u,v}}])$$

$$\overset{(u \notin S \text{ since } S \text{ is an IS})}{=} \sum_{v \in S} \sum_{e=(u,v) \in E} m^{2p_{u,v}} \cdot (1 - \sum_{z \in S: z \neq v} \mathbf{Pr}[X_z > m^{4p_{u,v}}])$$

$$= \sum_{v \in S} \sum_{e=(u,v) \in E} m^{2p_{u,v}} \cdot (1 - \sum_{z \in S: z \neq v} \sum_{m^{4p'} > m^{4p_{u,v}}} \mathbf{Pr}[X_z = m^{4p'}])$$

$$= \sum_{v \in S} \sum_{e=(u,v) \in E} m^{2p_{u,v}} \cdot (1 - \sum_{z \in S: z \neq v} \sum_{m^{4p'} > m^{4p_{u,v}}} \frac{1}{\sqrt{m^{4p'}}})$$

$$\geq \sum_{v \in S} \sum_{e=(u,v) \in E} m^{2p_{u,v}} \cdot (1 - \sum_{p' > p_{u,v}} \frac{1}{m^{2p'}})$$

$$\geq \sum_{v \in S} \sum_{e=(u,v) \in E} m^{2p_{u,v}} \cdot (1 - \frac{2}{m^{2(p_{u,v}+1)}})$$

$$= \mathsf{E}[\sum_{v \in S} X_v] - \sum_{v \in S} \sum_{e=(u,v) \in E} \frac{2}{m^2}$$

$$= \mathsf{E}[\sum_{v \in S} X_v] - \sum_{v \in S} |N(S)| \frac{2}{m^2}$$

$$\geq \mathsf{E}[\sum_{v \in S} X_v] - \frac{2}{m}.$$

**Soundness**  Say $G$ does not contain an independent set of size $k$. Let $S \subseteq V$ be a subset of vertices=random variables of size $|S| = k$. Let $E_1(S)$ denote the set of edges where one endpoint is in $S$, and likewise let $E_2(S)$ denote the set of edges where both endpoints are in $S$.

$$\mathsf{E}[\max_{v \in S} X_v] = \sum_{(u,v) \in E_1(S)} m^{4p_{u,v}} \cdot \mathbf{Pr}[X_v = m^{4p_{u,v}}] \cdot \mathbf{Pr}[\forall z \in S : X_z \leq m^{4p_{u,v}}]$$

$$+ 2 \sum_{(u,v) \in E_2(S)} m^{4p_{u,v}} \cdot \mathbf{Pr}[X_v = m^{4p_{u,v}} \text{ OR } X_u = m^{4p_{u,v}}] \cdot \mathbf{Pr}[\forall z \in S : X_z \leq m^{4p_{u,v}}]$$

$$\leq \sum_{(u,v) \in E_1(S)} m^{4p_{u,v}} \cdot \mathbf{Pr}[X_v = m^{4p_{u,v}}] + 2 \sum_{(u,v) \in E_2(S)} m^{4p_{u,v}} \cdot \mathbf{Pr}[X_v = m^{4p_{u,v}} \text{ OR } X_u = m^{4p_{u,v}}]$$

$$= \sum_{(u,v) \in E_1(S)} m^{2p_{u,v}} + \left( \sum_{(u,v) \in E_2(S)} 2m^{2p_{u,v}} - 1 \right)$$

$$= \mathsf{E}[\sum_{v \in S} X_v] - |E_2(S)|$$

$$\leq E[\sum_{v \in S} X_v] - 1.$$

$\square$

# B   Missing proofs from Section 3

## B.1   Preliminaries on submodular functions

Before we present the details of our algorithm, we prove (for completeness) the following well known property of submodular functions.

**Claim 1.** *Let $f(S)$ be a submodular function. Then,* $\max_{S:|S| \leq (1-\epsilon)k} f(S) \geq (1-\epsilon) \max_{S:|S| \leq k} f(S).$

*Proof.* Let $A = arg \max_{S:|S| \leq k} f(S)$ and $B \subseteq A, |B| = (1-\epsilon)k$ such that $f(B)$ is maximized among all subsets of $A$ of size $(1-\epsilon)k$. Let $\Delta(v|S)$ be the marginal contribution of an element $v$ to a set $S$, i.e. $\Delta(v|S) = f(S \cup \{v\}) - f(S)$.

$$f(A) = f(B) + \sum_{i=1}^{\epsilon k} \Delta(v_i | B \cup \{v_1, \ldots, v_{i=1}\})$$

$$\leq f(B) + \sum_{i=1}^{\epsilon k} \Delta(v_i | S)$$

$$\leq f(B) + \sum_{i=1}^{\epsilon k} \frac{f(B)}{|B|}$$

$$\leq f(B) + \sum_{i=1}^{\epsilon k} \frac{f(B)}{(1-\epsilon)k}$$

$$= f(B) + \epsilon k \frac{f(B)}{(1-\epsilon)k}$$

$$= \frac{1}{1-\epsilon} f(B)$$

$\square$

We also prove that the expected maximum is a submodular function.

**Fact 1.** $f(S) = \mathbb{E}[\max_{i \in S}\{X_i\}]$ *is a monotone submodular set function.*

*Proof.* Let $f_{\mathbf{v}}(S) = \max_{i \in S} \mathbf{v}_i$ be $f(S)$ restricted on an outcome $\mathbf{v}$. $f_{\mathbf{v}}$ is obviously submodular. For completeness, let $T$ be a subset of the random variables and $S$ a subset of $T$, and consider some $i \in [n] \setminus T$. $f_{\mathbf{v}}(S \cup \{i\}) - f_{\mathbf{v}}(S) = \max\{\max_{j \in S} \mathbf{v}_j, \mathbf{v}_i\} - \max_{j \in S} \mathbf{v}_j = \max\{\mathbf{v}_i - \max_{j \in S} \mathbf{v}_j, 0\} \geq \max\{\mathbf{v}_i - \max_{j \in T} \mathbf{v}_j, 0\} = f_{\mathbf{v}}(T \cup \{i\}) - f_{\mathbf{v}}(T)$. Finally, non-negative linear combinations of submodular functions are submodular, so $f(S) = \sum_{\mathbf{v}} \mathbf{Pr}[\mathbf{v}] f_{\mathbf{v}}(S)$ is submodular. $\square$

## B.2   PTAS and analysis

**Step 1: preprocessing far tails ($> \tau$).**   Let $\tau$ be the largest number such that $\exists S_\tau \subseteq [n], |S_\tau| = k$ that satisfies $\mathbf{Pr}[\max_{i \in S_\tau} X_i \geq \tau] = \epsilon$. For each $i \in [n]$, let $H_i := \mathsf{E}[X_i | X_i > \tau]$, and let $H_{max} := \max_{i \in [n]} H_i$. Let $\hat{X}_i$ be the random variable that takes value equal to $X_i$ if $X_i \leq \tau$, and if $X_i > \tau$, it takes either value $H_{max}$ w.p. $\frac{H_i}{H_{max}}$ or zero w.p. $1 - \frac{H_i}{H_{max}}$. Note that $\mathsf{E}[\hat{X}_i | \hat{X}_i > \tau] = H_i = \mathsf{E}[X_i | X_i > \tau]$.

**Claim 2.** *For every subset $S \subseteq [n], |S| = k$,*

$$\mathsf{E}[\max_{i \in S} \hat{X}_i] \in \left[(1-\epsilon)\mathsf{E}[\max_{i \in S} X_i], \frac{1}{(1-\epsilon)}\mathsf{E}[\max_{i \in S} X_i]\right].$$

*Proof.* Let $A$ be the event that $\max_{i \in S} X_i \leq \tau$. When $A$ occurs, $\max_{i \in S} \hat{X}_i = \max_{i \in S} X_i$.

Similarly, let $\neg A_i$ denote the event that $X_i > \tau$, and let $A_{-i}$ denote the event that all variables other than $i$ are below the threshold $\tau$. Note that $\neg A_i$ and $A_{-i}$ are independent, and conditioned on both, $\max_{j \in S} X_j = X_i$. Finally, we have that $\mathsf{E}[X_i | \neg A_i] = H_i = \mathsf{E}[\hat{X}_i | \neg A_i]$.

We now derive upper and lower bounds on $\mathsf{E}[\max_{i \in S} X_i | \neg A]$. The same proof will extend to $\hat{X}_i$. By inclusion-exlusion principle,

$$\mathbf{Pr}[\neg A]\mathsf{E}[\max_{i \in S} X_i | \neg A] = \sum_{\emptyset \neq U \subseteq S} \underbrace{\prod_{i \in U} \mathbf{Pr}[\neg A_i] \prod_{j \notin U} \mathbf{Pr}[A_j]}_{\mathbf{Pr}[U \text{ is the subset of variables that pass } \tau]} \mathsf{E}[\max_{i \in U} X_i | \forall i \in U \ X_i > \tau].$$

We now separate the cases of $|U| = 1$ and $|U| > 1$.

$$\mathbf{Pr}[\neg A]\mathsf{E}[\max_{i \in S} X_i | \neg A] = \sum_{i \in S} \underbrace{\mathbf{Pr}[\neg A_i]\,\mathbf{Pr}[A_{-i}]}_{\text{Only } X_i \text{ beats } \tau} \underbrace{\mathsf{E}[X_i | \neg A_i]}_{=H_i} + \sum_{\substack{U \subseteq S \\ |U| > 1}} (\dots).$$

Dropping the second term can only decrease the total, hence

$$\mathbf{Pr}[\neg A]\mathsf{E}[\max_{i \in S} X_i | \neg A] \geq \sum_{i \in S} \mathbf{Pr}[\neg A_i]\,\mathbf{Pr}[A_{-i}]H_i. \tag{1}$$

To obtain an upper bound on $\mathbf{Pr}[\neg A]\mathsf{E}[\max_{i \in S} X_i | \neg A]$, notice that conditioned on $\neg A$,

$$\max_{i \in S} X_i \leq \sum_{i \in S} X_i \mathbf{1}_{X_i > \tau}.$$

Hence, we have that

$$\mathbf{Pr}[\neg A]\mathsf{E}[\max_{i \in S} X_i | \neg A] \leq \mathbf{Pr}[\neg A]\mathsf{E}[\sum_{i \in S} X_i \mathbf{1}_{X_i > \tau} | \neg A]$$

$$= \sum_{i \in S} \mathbf{Pr}[\neg A]\mathsf{E}[X_i \mathbf{1}_{X_i > \tau} | \neg A]$$

$$= \sum_{i \in S} \underbrace{\mathsf{E}[X_i \mathbf{1}_{X_i > \tau}]}_{=H_i}. \tag{2}$$

Combining Eqs. (1) and (2), and recalling that they also hold for $\hat{X}_i$, we have that

$$\sum_{i \in S} \mathbf{Pr}[\neg A_i]\,\mathbf{Pr}[A_{-i}]H_i \leq \mathbf{Pr}[\neg A]\mathsf{E}[\max_{i \in S} X_i | \neg A], \mathbf{Pr}[\neg A]\mathsf{E}[\max_{i \in S} \hat{X}_i | \neg A] \leq \sum_{i \in S} \mathbf{Pr}[\neg A_i]H_i.$$

It's left to show that the lower and upper bound are close. Indeed, by the choice of $\tau$, $\mathbf{Pr}[A_{-i}] \geq \mathbf{Pr}[A] \geq 1 - \epsilon$. $\square$

**Step 2: discarding small values.** Now, let $W_i$ be the random variable that takes value equal to $\hat{X}_i$ when $\hat{X}_i \geq \epsilon^2 \tau$ and zero otherwise. The loss is, again, negligible.

**Claim 3.** $(1+\epsilon)OPT_{max}(W) \geq OPT_{max}(\hat{X})$.

*Proof.* First, notice that $OPT_{max}(W) \geq \epsilon\tau$, by the definition of $\tau$: for the subset $S_\tau$ such that $\mathbf{Pr}[\max_{i \in S_\tau} X_i \geq \tau] = \mathbf{Pr}[\max_{i \in S_\tau} W_i \geq \tau] = \epsilon$, so $\mathsf{E}[\max_{i \in S_\tau} W_i] \geq \mathbf{Pr}[\max_{i \in S_\tau} W_i \geq \tau]\tau \geq \epsilon\tau$. Second, consider the random variables $Z_i = \hat{X}_i - \epsilon^2 \tau$. Then $OPT_{max}(Z) = OPT_{max}(\hat{X}) - \epsilon^2 \tau$. Also, for all $i$, we can couple all outcomes of $W_i$ and $Z_i$ such that $W_i \geq Z_i$. Therefore, $OPT_{max}(W) \geq OPT_{max}(Z) = OPT_{max}(\hat{X}) - \epsilon^2 \tau \geq OPT_{max}(\hat{X}) - \epsilon OPT_{max}(W)$, which implies the lemma. $\square$

**Step 3: Interval and rounding.** We partition the range $[\epsilon^2\tau, \tau]$ into $\ell := \log_{1-\epsilon}(\epsilon^2) \in \tilde{O}(1/\epsilon)$ *intervals* $[\epsilon^2\tau, \tau] = \cup_{j=1}^{\ell} I_j$, where $I_j = [\frac{\epsilon^2}{(1-\epsilon)^{j-1}}\tau, \frac{\epsilon^2}{(1-\epsilon)^j}\tau)$. We round down the values within each interval. That is, let $Y_i$ be the random variable that takes value $\frac{\epsilon^2}{(1-\epsilon)^{j-1}}$ when $W_i$ takes value in $I_j$, for $j \in [\ell]$. Notice that this decreases the expected maximum by at most a $(1-\epsilon)$ factor.

**Observation 1.** *For all $S \subseteq [n]$, $\mathsf{E}[\max_{i \in S} Y_i] \geq (1-\epsilon)\mathsf{E}[\max_{i \in S} W_i]$.*

**Step 4: Core-Tail decomposition.** Let $\eta$ be the largest number such that there exists a set $S_\eta \subseteq [n], |S_\eta| = \epsilon k$, that satisfies

$$\mathbf{Pr}[\max_{i \in S_\eta} Y_i \geq \eta] = 1 - \epsilon. \tag{3}$$

We henceforth use $S_C$ to denote the set that attains equality in (3). We decompose each $Y_i$ into "core" ($C_i$) and "tail"($T_i$). Specifically, $C_i$ is the random variable that is equal $Y_i$ when $Y_i \leq \eta$ (and is zero otherwise), and $T_i$ is the random variable that is equal to $Y_i$ when $Y_i > \eta$ (and is zero otherwise).

We have used $\epsilon$-fraction of the budget to select the set $S_C$ (that is, $S_C$ has size $\epsilon k$). Next, we show that since the function $f(S) = \mathsf{E}[\max_{i \in S} Y_i]$ is a submodular function, using the remaining $(1-\epsilon)$-fraction of the budget to optimize contribution from tails recovers almost the same contribution as spending the entire budget on optimal tails. Let $S_T'$ be the subset of size $k'$ that is almost optimal with respect to $S_C$, i.e.

$$S_T' := \arg\max_{|S|=k'} \mathsf{E}\left[\max\left\{\max_{i \in S} T_i, \max_{i \in S_C} Y_i\right\}\right].$$

The following lemma compares the union of $S_T'$ and $S_C$ to the optimal set $S^*$ for the $Y_i$s.

**Lemma 3.**

$$\mathsf{E}\left[\max\left\{\max_{i \in S_T'} T_i, \max_{i \in S_C} Y_i\right\}\right] \geq (1 - O(\epsilon))\,\mathrm{OPT}_{max}(Y) \tag{4}$$

*Proof.* Let $S^*$ be the optimal set (of size $k$) for the $Y_i$s. We will show a slightly stronger bound, namely that the bound holds even when we replace the RHS of Eq. (4) with a union of $S^*$ and $S_C$:

$$\mathsf{E}\left[\max\left\{\max_{i \in S_T'} T_i, \max_{i \in S_C} Y_i\right\}\right] \geq (1 - O(\epsilon))\mathsf{E}\left[\max_{i \in S^* \cup S_C} Y_i\right]. \tag{5}$$

We first bound the loss to the RHS from truncating all the variables in $S^* \setminus S_C$ below $\eta$ (i.e. replacing those $Y_i$s with $T_i$s). Notice that the loss to the RHS of (5) is at most $\eta$ times the probability that none of the variables in $S_C$ exceed $\eta$. The latter, happens with probability at most $\epsilon$ by definition of $\eta$. Therefore,

$$\mathsf{E}\left[\max_{i \in S^* \cup S_C} Y_i\right] \leq \mathsf{E}\left[\max\left\{\max_{i \in S^*} T_i, \max_{i \in S_C} Y_i\right\}\right] + \epsilon\eta$$

$$\leq^{\text{(Eq. (3) + Markov's inequality)}} \mathsf{E}\left[\max\left\{\max_{i \in S^*} T_i, \max_{i \in S_C} Y_i\right\}\right] + \epsilon\frac{\mathsf{E}[\max_{i \in S_C} Y_i]}{1 - \epsilon}$$

$$\leq (1 + 2\epsilon)\mathsf{E}\left[\max\left\{\max_{i \in S^*} T_i, \max_{i \in S_C} Y_i\right\}\right]. \tag{6}$$

Now, if we let $S_T$ denote the optimal set of $k$ tails. By optimality of $S_T$, we have

$$\mathsf{E}\left[\max\left\{\max_{i \in S_T} T_i, \max_{i \in S_C} Y_i\right\}\right] \geq \mathsf{E}\left[\max\left\{\max_{i \in S^*} T_i, \max_{i \in S_C} Y_i\right\}\right]. \tag{7}$$

Finally, we use the fact that the expected max is a submodular function, to obtain:

$$\mathsf{E}\left[\max\left\{\max_{i \in S_T'} T_i, \max_{i \in S_C} Y_i\right\}\right] \geq (1 - \epsilon)\mathsf{E}\left[\max\left\{\max_{i \in S_T} T_i, \max_{i \in S_C} Y_i\right\}\right] \qquad \text{(Claim 1)}$$

$$\geq (1 - \epsilon)\mathsf{E}\left[\max\left\{\max_{i \in S^*} T_i, \max_{i \in S_C} Y_i\right\}\right] \qquad \text{(Eq. (7))}$$

$$\geq (1 - O(\epsilon))\mathsf{E}\left[\max_{i \in S^* \cup S_C} Y_i\right] \qquad \text{(Eq. (6))}.$$

$\square$

**Step 5: Discarding intervals with small relative contribution.** For a tail variable $T_i$ and interval $I \in \left\{[\epsilon^2\tau, \frac{\epsilon^2}{1-\epsilon}\tau), \ldots, [(1-\epsilon)\tau, (1-\epsilon)^2\tau), [(1-\epsilon)\tau, \tau], (\tau, \infty)\right\}$, recall that $T_i$ restricted to $I$ is a point mass: either one of the $\frac{\epsilon^2\tau}{(1-\epsilon)^{j-1}}$ or $H_{max}$. Let $T_i(I)$ denote its marginal contribution to the expectation, i.e. its probability times its value. We round down the probability on each point mass so its marginal contribution is equal to $(1 - \epsilon)^z \mathsf{E}[T_i]$ for some integer $z$.

For any set $|S| = k$, there is at most a constant probability that no variable passes $\eta$. To see this, consider partitioning $S$ into $1/\epsilon$ subsets $S_j$ of size $\epsilon k$.

$$\mathbf{Pr}[\max_{i \in S} Y_i < \eta] = \prod_j \mathbf{Pr}[\max_{i \in S_j} Y_i < \eta] \geq^{\text{(Eq. (3))}} \prod_j \epsilon = \epsilon^{1/\epsilon}.$$

Therefore, conditioned on being nonzero, any $T_i$ has a constant ($\geq \epsilon^{1/\epsilon}$) probability of attaining the maximum. Therefore

$$\mathsf{E}[\max_{i \in S'_T} Y_i] \geq \epsilon^{1/\epsilon} \sum_{i \in S'_T} \mathsf{E}[T_i]. \qquad (8)$$

We can therefore discard any intervals whose contribution is at most $\epsilon^{1/\epsilon + 3}\mathsf{E}[T_i]$, while reducing the expected max by at most a $(1 - \epsilon)$ factor. Formally, we have the following claim.

**Claim 4.** *Let $\hat{T}_i$ denote the modified variable $T_i$ where we discard intervals with contribution at most $\epsilon^{1/\epsilon + 3}\mathsf{E}[T_i]$, and otherwise round it down to the nearest power of $(1 - \epsilon)$. Then,*

$$\mathsf{E}\left[\max\left\{\max_{i \in S'_T} \hat{T}_i, \max_{i \in S_C} Y_i\right\}\right] \geq (1 - \epsilon)^2\mathsf{E}\left[\max\left\{\max_{i \in S'_T} T_i, \max_{i \in S_C} Y_i\right\}\right].$$

*Proof.* Let $\hat{\hat{T}}_i$ denote the modified variable where we only discard intervals with low contributions, but without the rounding. Discarding each interval with contribution at most $\epsilon^{1/\epsilon + 3}\mathsf{E}[T_i]$ can decrease the expected max by at most $\epsilon^{1/\epsilon + 3}\mathsf{E}[T_i]$. There are $\tilde{O}(1/\epsilon) < 1/\epsilon^2$ intervals, hence discarding all low-contribution intervals for variable $i$ decreases the expected max by $\epsilon^{1/\epsilon + 3}\mathsf{E}[T_i]$. Summing accross all intervals, we have that

$$\mathsf{E}\left[\max\left\{\max_{i \in S'_T} \hat{\hat{T}}_i, \max_{i \in S_C} Y_i\right\}\right] \geq \mathsf{E}\left[\max\left\{\max_{i \in S'_T} T_i, \max_{i \in S_C} Y_i\right\}\right] - \epsilon^{1/\epsilon + 3} \sum_{i \in S'_T} \mathsf{E}[T_i]$$

$$\geq^{\text{(Eq. (8))}} (1 - \epsilon)\mathsf{E}\left[\max\left\{\max_{i \in S'_T} T_i, \max_{i \in S_C} Y_i\right\}\right].$$

Rounding down each contribution to the nearest power of $(1 - \epsilon)$ can cost at most another factor of $1 - \epsilon$. $\square$

**The brute-force algorithm** Notice that for each interval $I$, the marginal contribution from the point mass of $\hat{T}_i$ takes one of $1/\epsilon + 3$ values. Therefore, each variable belongs to one of $(1/\epsilon)^{\tilde{O}(1/\epsilon)} = O(1)$ many *types*, which describe the relative marginal contribution from each point mass.

We now guess an approximate *type-histogram*, i.e. the optimal number of variables for each type, rounded down to the nearest power of $(1 + \epsilon)$. There are $O(\log(k))$ possible guesses for each type, so $\log^{(1/\epsilon)^{\tilde{O}(1/\epsilon)}}(k)$ possible type-histograms total. For each type-histogram, we generate a candidate set of $\leq k$ variables by taking, for each type, the variables from that type with maximal $\mathsf{E}[T_i]$. We estimate the expected maximum of each candidate set, and return the best one across all type-histograms.

**Running time** The five pre-processing steps run in linear time (aka linear in sum of variable supports). The brute-force algorithm runs in time $O(k \cdot \text{polylog}(k))$.

## C Proof of Theorem 3

We reduce from the Densest-$\kappa$-subgraph problem, formally defined as follows.

**Definition 2** (Densest $\kappa$-subgraph ($D\kappa S$))**.** *We are given a $n$-vertex graph $G = (V, E)$ and an integer $\kappa$. The goal is to select a subgraph of $G$ of size $\kappa$ with maximum density (average degree).*

[AAM$^+$11] prove the following hardness result.

**Theorem 5** ([AAM$^+$11])**.** *If there is no polynomial time algorithm for solving the hidden clique problem for a planted clique of size $n^{1/3}$ in the random graph $G(n, 1/2)$, then for any $2/3 \geq \epsilon > 0$, $\delta > 0$, there is no polynomial time algorithm that distinguishes between a graph $G$ on $N$ vertices containing a clique of size $\kappa = N^{1-\epsilon}$, and a graph $G'$ on $N$ vertices in which the densest subgraph on $\kappa$ vertices has density at most $\delta$.*

[Man17] proves the following hardness result.

**Theorem 6** ([Man17])**.** *There is a constant $c > 0$ such that, assuming the exponential time hypothesis, no polynomial-time algorithm can, given a graph $G$ on $n$ vertices and a positive integer $\kappa \leq n$, distinguish between the following two cases:*

- *There exist $\kappa$ vertices of $G$ that induce the $\kappa$-clique.*

- *Every $\kappa$-subgraph of $G$ has density at most $n^{-1/(\log \log n)^c}$*

In the following, we show that given a graph $G$ on $n$ vertices, we can construct $n$ random variables, $X_1, \ldots, X_n$ such that

- (Completeness) If there exists a subset of vertices $S^*$ of size $k$ such that $|E \cap (S^* \times S^*)| = \ell$, then there exists a subset of random variables whose expected second largest value is at least $\ell$.

- (Soundness) If for all subsets $S$ of size $k$ $|E \cap (S \times S)| < \ell$, then there is no subset of random variables whose expected second largest value is more than $\ell + 1/2k$.

Combining with Theorems 5 and 6, Theorem 3 follows immediately.

Our construction works as follows. Given a graph $G = (V, E)$, we make a random variable $X_i$ for each vertex $i \in V$. If $(i, j) \in E$, then we add the value $p_{ij}^2$ with probability $1/p_{ij}$ to the support of $X_i$ and $X_j$, where $p_{ij} = \frac{1}{(2k+1)^{\pi(i,j)}}$, where $\pi : E \to \mathbb{N}$ is an arbitrary ordering of the edges (concretely, one can take $\pi(i, j) = \frac{(\max\{i,j\}-1)(\max\{i,j\}-2)}{2} + \min\{i, j\}$).

**Completeness.** Assume that there exists $S^* \subseteq V$, $|S^*| = k$, such that $|E \cap (S^* \times S^*)| = \ell$. Then

$$
\begin{aligned}
\mathsf{E}[\text{smax}_{i \in S^*} X_i] &= \sum_v v \cdot \mathbf{Pr}[\text{smax}_{z \in S} X_z = v] \\
&= \sum_{i,j \in S^*:(i,j)\in E} p_{ij}^2 \cdot \mathbf{Pr}[\text{smax}_{z \in S} X_z = p_{ij}^2] \\
&\geq \sum_{i,j \in S^*:(i,j)\in E} p_{ij}^2 \cdot \mathbf{Pr}[X_i = X_j = p_{ij}^2] \\
&= \sum_{i,j \in S^*:(i,j)\in E} p_{ij}^2 \frac{1}{p_{ij}^2} \\
&= \ell.
\end{aligned}
$$

**Soundness.** Let $S$ be an arbitrary subset of random variables. We'll show that $\mathsf{E}[\text{smax}_{i \in S} X_i]$ is at most $\ell + 1/2k$, where $\ell$ is the number of edges between vertices of $S$.

$$\mathsf{E}[\mathrm{smax}_{i \in S} X_i] = \sum_{i,j \in S^*:(i,j) \in E} p_{ij}^2 \cdot \mathbf{Pr}[\mathrm{smax}_{z \in S} X_z = p_{ij}^2]$$

$$= \sum_{i,j \in S^*:(i,j) \in E} p_{ij}^2 \cdot (\mathbf{Pr}[\mathrm{smax}_{z \in S} X_z = \max_{z \in S} X_z = p_{ij}^2]$$

$$+ \mathbf{Pr}[\mathrm{smax}_{z \in S} X_z = p_{ij}^2 \ \& \ \max_{z \in S} X_z > p_{ij}^2]).$$

Since $X_i$ and $X_j$ are the only variables that can take value $p_{ij}^2$ we have that

$$\mathbf{Pr}[\mathrm{smax}_{z \in S} X_z = \max_{z \in S} X_z = p_{ij}^2] \le \mathbf{Pr}[X_i = X_j = p_{ij}^2] = \frac{1}{p_{ij}^2}. \tag{9}$$

For the same reason, the probability that the second largest value is $p_{ij}^2$ and the maximum value is at least $p_{ij}^2$ is upper bounded by $\frac{2}{p_{ij}} \sum_{k \in S^*} \mathbf{Pr}[X_k > p_{ij}^2]$. We have that

$$\mathbf{Pr}[X_k > p_{ij}^2] = \sum_{v > p_{ij}^2} \mathbf{Pr}[X_k = v]$$

$$= \sum_{z > \pi(i,j)} \mathbf{Pr}[X_k = p_z^2]$$

$$\le \sum_{z = \pi(i,j)+1}^{\infty} \frac{1}{(2k+1)^z}$$

$$= \frac{1}{2k} \cdot \frac{1}{(2k+1)^{\pi(i,j)}}$$

$$= \frac{1}{2kp_{ij}^2}$$

Thus

$$\mathbf{Pr}[\mathrm{smax}_{z \in S} X_z = p_{ij}^2 \ \& \ \max_{z \in S} X_z > p_{ij}^2] \le \frac{2}{p_{ij}} \cdot k \cdot \frac{1}{2kp_{ij}^2} = \frac{1}{p_{ij}^3} \tag{10}$$

Plugging in (9) and (10) we get that

$$\mathsf{E}[\mathrm{smax}_{i \in S} X_i] \le \sum_{i,j \in S^*:(i,j) \in E} p_{ij}^2 \cdot (\frac{1}{p_{ij}^2} + \frac{1}{p_{ij}^3})$$

$$= \sum_{i,j \in S^*:(i,j) \in E} 1 + \frac{1}{p_{ij}}$$

$$= \ell + \sum_{i,j \in S^*:(i,j) \in E} \frac{1}{(2k+1)^{\pi(i,j)}}$$

$$\le \ell + \sum_{z=1}^{\infty} \frac{1}{(2k+1)^z}$$

$$= \ell + \frac{1}{2k}. \quad \square$$

## D  Proofs missing from Section 5

### D.1  Proof of Lemma 1

*Proof.* First, we lower bound the probability that the maximum is at least the smallest $\alpha_p^{(i)}$. Let $\alpha_{min} = \min_{i \in S} \alpha_p^{(i)}$. For all $i \in S$, $\mathbf{Pr}[X_i \le \alpha_{min}] \le \mathbf{Pr}[X_i \le \alpha_p^{(i)}] = 1 - 1/p$. Therefore,

$\mathbf{Pr}[\max_{i\in S}\hat{X}_i \leq \alpha_{min}] = \mathbf{Pr}[\forall_{i\in S}\hat{X}_i \leq \alpha_{min}] \leq (1-1/p)^k$. Thus, $\mathbf{Pr}[\max_{i\in S}\hat{X}_i \geq \alpha_{min}] \geq 1-(1-1/p)^k$. Second, conditioned on the maximum value being at least $\alpha_{min}$, our algorithm has, in fact, picked the random variable that takes the largest value (out of all $n$ random variables). To see this most clearly, notice that the only (truncated) random variables that can take values at least $\alpha_{min}$ are in $S$. The first part of the lemma follows from combining the two observations.

For the second part of the lemma, notice that $\mathbf{Pr}[\mathrm{smax}_{i\in S}\hat{X}_i \leq \alpha_{min}] = \mathbf{Pr}[\text{for all } i \in S, \hat{X}_i \leq \alpha_{min}] + \mathbf{Pr}[\text{for all but one } i \in S, \hat{X}_i \leq \alpha_{min}]$. The first term is at most $(1-1/p)^k$. The second term is at most $\sum_{j\in S}\mathbf{Pr}[\hat{X}_j \geq \alpha_{min}]\prod_{i\neq j\in S}\mathbf{Pr}[\hat{X}_i < \alpha_{min}] \leq k(1-1/p)^{k-1}$. So, overall, $\mathbf{Pr}[\mathrm{smax}_{i\in S}\hat{X}_i \leq \alpha_{min}] \leq (k+1)(1-1/p)^{k-1}$, and thus $\mathbf{Pr}[\mathrm{smax}_{i\in S}\hat{X}_i \geq \alpha_{min}] \geq 1-(k+1)(1-1/p)^{k-1}$. When this event occurs, the selected subset of variables includes all random variables whose have value at least $\alpha_{min}$; the second part of the lemma follows. $\square$

## D.2 Proofs missing from Section 5.2

Let $Con[X \geq x] = \mathsf{E}[X|X \geq x]\cdot Pr[X \geq x] = \int_x^\infty zf(z)dz$. Let $G_t$ be the set of random variables that got eliminated in round $t$ of Algorithm 1, for $t = 0,\ldots,\log_2 k - 1$, i.e. $G_t = Q_t \setminus Q_{t+1}$, and let $G_{\log_2 k} = Q_{\log_2 k}$, i.e. the unique random variable that survived the first $\log_2 k - 1$ rounds.

### D.2.1 Upper bounding the tail

The upper bounds on the tail contribution will hold only for MHR random variables. We first bound the contribution above $\beta_1$ for the sum of all random variables except $G_{\log_2 k}$ in Lemma 4. We then proceed to bound the contribution above $\beta_2$ for $G_{\log_2 k}$, in Lemma 5. When upper bounding the tail of the expected maximum we need both lemmas, but for the case of the expected second highest value, we can safely exclude one random variable.

**Lemma 4.** *Let $X_1,\ldots,X_k$ be MHR random variables. For all $i \in [k] \setminus G_{\log_2 k}$ and $\epsilon \in (0,1/16)$, let $S_i = Con[X_i \geq \log_2(1/\epsilon)\beta_1]$. Then $\sum_{i\in[k]\setminus G_{\log_2 k}} S_i \leq 8\sqrt{\epsilon}\log_2(1/\epsilon)\beta_1$.*

*Proof.* Let $d = \log_2(1/\epsilon)$, and notice that $d > 4$ since $\epsilon < 1/16$.

For $i \in G_t$, we know that $\alpha^{(i)}_{\sqrt{k/2^t}} \leq \beta_t$. Furthermore, by Lemma 6, $d\alpha^{(i)}_{\sqrt{k/2^t}} \geq \alpha^{(i)}_{(\sqrt{k/2^t})^d}$. Therefore, we have that $Con[X_i \geq d\beta_t] \leq Con[X_i \geq d\alpha^{(i)}_{\sqrt{k/2^t}}] \leq Con[X_i \geq \alpha^{(i)}_{(\sqrt{k/2^t})^d}]$.

Using Lemma 7 we get

$$Con[X_i \geq \alpha^{(i)}_{(\sqrt{k/2^t})^d}] \leq 6\alpha^{(i)}_{(\sqrt{k/2^t})^d}(\sqrt{2^t/k})^d \leq 6d\beta_t(\sqrt{2^t/k})^d.$$

Since $|G_t| = k/2^{t+1}$, we have

$$\sum_{i\in G_t} S_i \leq 6d\beta_t(\sqrt{2^t/k})^d \cdot k/2^{t+1} = \frac{3d\beta_t}{\sqrt{k}^{d-2}} \cdot (\sqrt{2}^t)^{d-2}$$

Therefore, the total contribution to the tail is

$$\sum_{i\in[k]\setminus G_{\log_2 k}} S_i \leq \sum_{t=0}^{\log_2 k - 1} \frac{3d\beta_t}{\sqrt{k}^{d-2}} \cdot (\sqrt{2}^t)^{d-2}$$

$$\leq \frac{3d\beta_1}{\sqrt{k}^{d-2}} \sum_{t=0}^{\log_2 k - 1} (\sqrt{2}^{d-2})^t$$

$$= \frac{3d\beta_1}{\sqrt{k}^{d-2}} \cdot \frac{(\sqrt{2}^{d-2})^{\log_2 k} - 1}{\sqrt{2}^{d-2} - 1}$$

$$= \frac{3d\beta_1}{\sqrt{k}^{d-2}} \cdot \frac{(\sqrt{k})^{d-2} - 1}{\sqrt{2}^{d-2} - 1}$$

$$\leq \frac{3d\beta_1}{\sqrt{2}^{d-2} - 1}$$

$$\leq \frac{8d\beta_1}{\sqrt{2}^d} = 8\sqrt{\epsilon}\log_2(1/\epsilon)\beta_1$$

where we used the fact that $\frac{2^{d+2}}{2^d-1} \leq \frac{8}{3}$ for $d \geq 4$. $\qquad\square$

**Lemma 5.** *Let $i$ be the unique element in $G_{\log_2 k}$. Then, if $X_i$ is MHR*

$$Con[X_i \geq \log_2(1/\epsilon)\beta_2] \leq 6\sqrt{\epsilon}\log_2(1/\epsilon)\beta_2, \quad \text{for all } \epsilon \in (0, 1/16).$$

*Proof.* Let $d = \log_2(1/\epsilon)$.

$$Con[X_i \geq d\beta_2] = Con[X_i \geq d\alpha^{(i)}_{\sqrt{2}}]$$

$$\leq^{(Lemma\ 6)} Con[X_i \geq \alpha^{(i)}_{\sqrt{2}^d}]$$

$$\leq^{(Lemma\ 7)} \frac{6\alpha^{(i)}_{\sqrt{2}^d}}{\sqrt{2}^d}$$

$$\leq^{(Lemma\ 6)} 6\sqrt{\epsilon}\log_2(1/\epsilon)\beta_2,$$

where in the application of Lemma 7 we used the fact that $\sqrt{2}^d = \sqrt{2}^{\log_2(1/\epsilon)} \geq 2$ for $\epsilon < 1/16$. $\qquad\square$

The maximum of $k$ random variables is upper bounded by their sum, therefore by combining Lemmas 4 and 5 we get the following corollary.

**Corollary 1.** *Let $X_1, \ldots, X_k$ be MHR random variables and $\beta = \max\{\beta_1, \beta_2\}$ be the maximum of the two values output by Algorithm 1. Then for all $\epsilon \in (0, 1/16)$ we have*

$$\int_{\beta \log_2(1/\epsilon)}^{\infty} x f_{max}(x) dx \leq 14\sqrt{\epsilon}\log_2(1/\epsilon)\beta,$$

*where $f_{max}(x)$ is the probability density function of the random variable $\max_{i \in [k]} X_i$.*

For the second largest value, observe that its expected value is upper bounded by the sum of the random variables minus (any) one of them. Therefore, by excluding the random variable in $G_{\log_2 k}$, we can upper bound the tail of the expected second highest value using Lemma 4.

**Corollary 2.** *Let $X_1, \ldots, X_k$ be MHR random variables and $\beta_1$ be the first output of Algorithm 1. Then for all $\epsilon \in (0, 1/16)$ we have*

$$\int_{\beta_1 \log_2(1/\epsilon)}^{\infty} x f_{\text{smax}}(x) dx \leq 8\sqrt{\epsilon}\log_2(1/\epsilon)\beta_1,$$

*where $f_{\text{smax}}(x)$ is the probability density function of the random variable $\text{smax}_{i \in [k]} X_i$.*

### D.2.2 Lower bounding the probability of being in the tail

We will repeatedly use the following two facts.

**Lemma 6** ([CD15]). *Let $X$ be an MHR random variable. Then for $p \geq 1$ and $d \geq 1$, $d\alpha_p \geq \alpha_{p^d}$.*

**Lemma 7** ([CD15]). *Let $X$ be an MHR random variable. Then for all $p \geq 2$, $Con[X \geq \alpha_p] \leq 6\alpha_p/p$.*

We now lower bound the probability that the largest and second largest value are above the outputs $\max\{\beta_1, \beta_2\}$ and $\beta_1$ of Algorithm 1. These bounds hold even if the variables are not MHR.

**Lemma 8.** *For any random variables (possibly not MHR) $X_1, \ldots, X_k$ the threshold $\beta = \max\{\beta_1, \beta_2\}$ given by Algorithm 1 satisfies $\mathbf{Pr}[\max_i X_i \geq \beta] \geq 1/2$.*

*Proof.* For the unique element $i$ in $G_{\log_2 k}$ we have that $\mathbf{Pr}[X_i \geq \beta_2] = 1/\sqrt{2} \geq 1/2$; this covers the case that $\max\{\beta_1, \beta_2\} = \beta_2$. For the case that $\max\{\beta_1, \beta_2\} = \beta_1$ we prove that for all $t = 0, \ldots, \log_2 k - 1$, $\mathbf{Pr}[\max_i X_i \geq \beta_t] \geq 1/2$. This is sufficient, since $\beta_1 = \max_{t=0,\ldots,\log_2 k - 1} \beta_t$.

Notice that, for all $i$ that survived round $t$, i.e. $Q_{t+1}$, we have that $\alpha^{(i)}_{\sqrt{k/2^t}} \geq \beta_t$. Therefore, for those random variables, $\mathbf{Pr}[X_i \leq \beta_t] \leq \mathbf{Pr}[X_i \leq \alpha^{(i)}_{\sqrt{k/2^t}}] = 1 - \sqrt{\frac{2^t}{k}}$. $|Q_{t+1}| = k/2^{t+1}$, so we get

$$\mathbf{Pr}[\max_i X_i \geq \beta_t] \geq \mathbf{Pr}[\exists i \in Q_{t+1} : X_i \geq \beta_t] \geq 1 - (1 - \frac{\sqrt{2^t}}{\sqrt{k}})^{k/2^{t+1}}$$

$$\geq 1 - \left(\frac{1}{\sqrt{e}}\right)^{\sqrt{k}/\sqrt{2^t}} \geq 1 - \left(\frac{1}{\sqrt{e}}\right)^{\sqrt{k}/\sqrt{2^{\log_2 k - 1}}} \geq 1 - \left(\frac{1}{\sqrt{e}}\right)^{\sqrt{2}} \geq 1/2. \qquad \square$$

**Lemma 9.** *For any random variables (possibly not MHR) $X_1, \ldots, X_k$ the value $\beta_1$ given by Algorithm 1 satisfies $\mathbf{Pr}[\mathrm{smax}_i X_i \geq \beta_1] \geq 0.098$.*

*Proof.* We prove that for all $t = 0, \ldots, \log_2 k - 1$, $\mathbf{Pr}[\mathrm{smax}_i X_i \geq \beta_t] \geq 0.098$, which suffices since $\beta_1 = \max_{t=0,\ldots,\log_2 k-1} \beta_t$. In round $t$, for all $k/2^{t+1}$ surviving random variables $X_i$, as well as the eliminated random variable $X_\ell$ with the largest (among eliminated random variables) $\alpha_{\sqrt{k/2^t}}$, we have that $\alpha^{(i)}_{\sqrt{k/2^t}} \geq \beta_t$. Therefore, for all $i \in Q_{t+1} \cup \{\ell\}$, $\mathbf{Pr}[X_i \leq \beta_t] \leq 1 - \frac{\sqrt{2^t}}{\sqrt{k}}$, with equality for the random variable $X_\ell$.

$$\mathbf{Pr}[\mathrm{smax}_i X_i \geq \beta_t] \geq \mathbf{Pr}[\mathrm{smax}_{i \in Q_{t+1} \cup \{\ell\}} X_i \geq \beta_t]$$

$$= 1 - (\mathbf{Pr}[\max_{i \in Q_{t+1} \cup \{\ell\}} X_i \leq \beta_t] + \mathbf{Pr}[\max_{i \in Q_{t+1} \cup \{\ell\}} X_i \geq \beta_t \text{ and } \mathrm{smax}_{i \in Q_{t+1} \cup \{\ell\}} X_i \leq \beta_t])$$

$$\geq 1 - \prod_{i \in Q_{t+1} \cup \{\ell\}} \mathbf{Pr}[X_i \leq \beta_t] - \sum_{i \in Q_{t+1}} \mathbf{Pr}[X_i \geq \beta_t] \prod_{j \neq i} \mathbf{Pr}[X_j \leq \beta_t]$$

$$\quad - \mathbf{Pr}[X_\ell \geq \beta_t] \prod_{j \neq \ell} \mathbf{Pr}[X_j \leq \beta_t]$$

$$\geq 1 - \left(1 - \frac{\sqrt{2^t}}{\sqrt{k}}\right)^{k/2^{t+1}+1} - \frac{k}{2^{t+1}} \cdot \left(1 - \frac{\sqrt{2^t}}{\sqrt{k}}\right)^{k/2^{t+1}} - \frac{\sqrt{2^t}}{\sqrt{k}} \left(1 - \frac{\sqrt{2^t}}{\sqrt{k}}\right)^{k/2^{t+1}}$$

$$= 1 - \left(\frac{k}{2^{t+1}} + 1\right) \left(1 - \frac{\sqrt{2^t}}{\sqrt{k}}\right)^{k/2^{t+1}}.$$

When $\frac{k}{2^{t+1}}$ takes small values the standard approximation $(1 - 1/n)^n \leq 1/e$ is not good enough. We take cases. When $\frac{k}{2^{t+1}} \geq 32$ we use the standard exponential approximation, and argue that the minimum of the resulting function is at least 0.395. When $\frac{k}{2^{t+1}} < 32$, i.e. when it takes the values $1, 2, 4, 8$ and $16$ we simply compute the value of the function above; the smallest of the five is 0.098 when $\frac{k}{2^{t+1}} = 8$.

Let $x = \log_2(\frac{k}{2^{t+1}})$. Our goal is to find the minimum of $g(x) = 1 - (2^x + 1)\left(1 - \frac{1}{\sqrt{2^{x+1}}}\right)^{2^x}$. The standard approximation $(1 - 1/n)^n \leq 1/e$ is not good enough for small values of $x$ (specifically $x \geq$

For $x \geq 5$, we use the standard approximation $(1 - 1/n)^n \leq 1/e$:

$$1 - (2^x + 1)\left(1 - \frac{1}{\sqrt{2^{x+1}}}\right)^{2^x} \geq 1 - (2^x + 1)\left(\frac{1}{\sqrt{e}}\right)^{\sqrt{2^{x+1}}} = f(x)$$

Taking the derivative, we have that $f'(x) = \frac{1}{4}\ln(2)e^{-\sqrt{2^{x-1}}}\left(2^{x+2} - \sqrt{2^{x+1}}(2^x + 1)\right)$; this expression is negative when $2^{x+2} \leq \sqrt{2^{x+1}}(2^x + 1)$, which holds for $x \geq 5$. Therefore, for $x \geq 5$, $f(x)$ achieves its maximum at $x = 5$, where it takes the value $f(5) = 1 - \frac{33}{e^4} \geq 0.395$.

Therefore, it remains to confirm the lower bound on $g(x) = 1 - (2^x + 1)\left(1 - \frac{1}{\sqrt{2^{x+1}}}\right)^{2^x}$ for the cases of $x = 0, \ldots, 4$ (equivalently, $\frac{k}{2^{t+1}} = 1, 2, 4, 8$ and $16$) where we have:

- $g(0) = \sqrt{2} - 1 \approx 0.412$.

- $g(1) = 1/4$.

- $g(2) = 3/64(-167 + 120sqrt(2)) \approx 0.126$.

- $g(3) = 6487/65536 \geq 0.098$.

- $g(4) = 1 - 17(1 - 1/(4sqrt(2)))^16 \approx 0.243$

The lowest number is $g(3)$, therefore, $\mathbf{Pr}[\text{smax}_i X_i \geq \beta_t] \geq 0.098$. Since $\max_{t=0,\log_2 k-1} \beta_t = \beta_1$, we get $\mathbf{Pr}[\text{smax}_i X_i \geq \beta_1] \geq 0.098$. $\qquad\square$

### D.2.3  Bounding the loss of truncation

Here, we prove our main bound on the loss from truncating.

*Proof of Lemma 2.* Observe that since Algorithm 1 only uses top quantiles smaller than $\frac{1}{\sqrt{k}}$, then the outputs $(\beta_1, \beta_2)$ of Algorithm 1 with inputs the $X_i$s are identical to its outputs with inputs the $\hat{X}_i$s. With this observation at hand we can proceed as follows. Let $\beta = \max\{\beta_1, \beta_2\}$. First, combining Markov's inequality with Lemmas 8 and 9[4] we get the following inequalities

$$\frac{\mathsf{E}[\max_i \hat{X}_i]}{\beta} \geq \mathbf{Pr}[\max_i \hat{X}_i \geq \beta] \geq 1/2 \tag{11}$$

$$\frac{\mathsf{E}[\text{smax}_i \hat{X}_i]}{\beta_1} \geq \mathbf{Pr}[\text{smax}_i \hat{X}_i \geq \beta_1] \geq 0.098 \tag{12}$$

For $\mathsf{E}[\max_i X_i]$ we have:

$$\mathsf{E}[\max_i X_i] = \int_0^{\log_2(1/\epsilon)\beta} x f_{max}(x)dx + \int_{\log_2(1/\epsilon)\beta}^{\infty} x f_{max}(x)dx$$
$$\leq^{(Corollary\ 1)} \beta(\log_2(1/\epsilon) + 14\sqrt{\epsilon}\log_2(1/\epsilon))$$
$$\leq^{(Eq\ (11))} 2(\log_2(1/\epsilon) + 14\sqrt{\epsilon}\log_2(1/\epsilon))\mathsf{E}[\max_i \hat{X}_i].$$

By picking $\epsilon = 0.00075$ we have $\mathsf{E}[\max_i X_i] \leq 28.8\mathsf{E}[\max_i \hat{X}_i]$.

For $\mathsf{E}[\text{smax}_i X_i]$ we have:

$$\mathsf{E}[\text{smax}_i X_i] = \int_0^{\log_2(1/\epsilon)\beta_1} x f_{smax}(x)dx + \int_{\log_2(1/\epsilon)\beta_1}^{\infty} x f_{smax}(x)dx$$
$$\leq^{(Corollary\ 2)} \beta_1(\log_2(1/\epsilon) + 8\sqrt{\epsilon}\log_2(1/\epsilon))$$
$$\leq^{(Eq\ (12))} \frac{1}{0.098}(\log_2(1/\epsilon) + 8\sqrt{\epsilon}\log_2(1/\epsilon))\mathsf{E}[\text{smax}_i \hat{X}_i].$$

By picking $\epsilon = 0.0074$ we have $\mathsf{E}[\text{smax}_i X_i] \leq 122\mathsf{E}[\text{smax}_i \hat{X}_i]$. $\qquad\square$

# E  Additional Experimental Results

**Additional Implementation Details.**  We implemented all our algorithms in Tensorflow, on Google's Colab.

For synthetic data, a random variable $X_i$ is constructed by first sampling a mean $\mu_i$ from $U[0, 60]$ and a $\sigma_i$ from $U[0, 30]$, and taking the empirical over 5000 samples from a Normal distribution $\mathcal{N}_i(\mu_i, \sigma_i)$, where the sampled values were rounded up to 0 and down to $V_{max} = 1000$ if outside of the $[0, V_{max}]$ range. An experiment constructs $n = 500$ random variables, and selects a subset of size $k$, for $k = 10, 20$ and 30 for each of the different methods. For a selected subset $S$, we compute $\mathsf{E}[\max_{i \in S} X_i]$ and $\mathsf{E}[\text{smax}_{i \in S} X_i]$, which is the "score" for that experiment. We ran 100 experiments.

For the small versus big data experiments, we have a similar setup. A random variable $X_i$ is constructed by first sampling a mean $\mu_i$ from $U[0, 60]$ and a $\sigma_i$ from $U[0, 30]$ and a uniformly random label $\{s, b\}$. If the label is $s$ $X_i$ is the empirical over 10 samples from $\mathcal{N}_i(\mu_i, \sigma_i)$, otherwise it is the empirical over 5000 samples. Once a method selects a subset $S$, its score (true performance) is $\mathsf{E}[\max_{i \in S} \mathcal{N}_i(\mu_i, \sigma_i)]$ and $\mathsf{E}[\text{smax}_{i \in S} \mathcal{N}_i(\mu_i, \sigma_i)]$, which is computed via sampling. Specifically, we sample 500 times from each $\mathcal{N}_i(\mu_i, \sigma_i)$, $i \in S$, remembering the largest/second largest value, and then take the average. Our plots show the number of small data candidates selected versus the true performance.

For the Twitter data, we have a dataset of 8 million tweets. The first 2 million tweets (in chronological order) are used for feature collection. We drop all entries with fewer than 5 likes and pre-process the text, removing stopwords ("this", "and", etc) and stemming (reducing words to their root, e.g. "jumped", "jumping" get mapped to "jump"). We find the set of distinct words in this set, and out of those, use as features the ones that appear at least 10 and at most 350 times. The purpose of the upper bound is (1) to not take into account words like "BTC" that appear in almost all tweets, (2) keep the number of features small enough for computation to be feasible. We get 5500 features that we use to train our models in the next 2 million tweets (without dropping any entries like in the feature collection). Regression is the simplest to train. For Quantile, we train a neural network (the framework we used is Keras) with 2 hidden layers, with quantile loss, at quantiles $[0.7, 0.8, 0.9, 0.95, 0.99]$. This is the final model for the Quantile method. The KR method uses these models to filter out the train set (the same 2 million entries quantile was trained on) by dropping all entries with fewer "likes" than the quantile prediction. We train with squared loss on the remaining entries. This concludes the training step. For the final step, we randomly perturb the last 4 million tweets and split it into non-overlapping chunks of $n = 500$ tweets. One experiment samples a chunk, and picks, for each method, a subset of size $k = 10, 20$ and 30 by ranking the entries based on the value of the prediction. The score of a method for this experiment is the true largest/second largest number of "likes" in the set picked. We do 8000 experiments.

**Additional Figures**  Figure 4 shows the results for the second largest objective, on the synthetic data. Figure 5 shows the results for the second largest objective on the Twitter data.

In Figures 6 and 7 we have the percentage of small data candidates and expected maximum for the selected set, for the Quantile and KR algorithm (respectively), for different quantiles.

(a) $k = 10$　　　　　　　　　　　　　　(b) $k = 30$

Figure 4: Comparing the average performance (errors bars show standard deviation divided by square root of number of experiments) of the score-based algorithms and Greedy, for selecting $k$ out of $n = 500$ distributions, for the expected second largest value objective.

(a) $k = 10$　　　　　　　　　　　　　　(b) $k = 30$

Figure 5: Comparing the average performance (errors bars show standard deviation divided by square root of number of experiments) of the KR, quantile and regression methods, for selecting $k$ out of $n = 500$ distributions, for the expected second largest value objective.

Figure 6: Percentage of small data candidates and expected maximum for the quantile algorithm, for different quantiles.

Figure 7: Percentage of small data candidates and expected maximum for the KR algorithm, for different quantiles.

## Footnotes

[4]Note that these lemmas do not need the random variables to be MHR, which is important since distributions with point masses, like the $\hat{X}_i$s, are not MHR, as $\log(1 - F_{\hat{X}_i}(x))$ is not concave.