[Reviews · NeurIPS 2020]

Review 1

Summary and Contributions: We are given n independent random variables with known distributions. We want to select k of them so as to maximize the expected maximum of the k random outcomes. The paper constructs algorithms for this task, shows simulations, and considers the more challenging case of maximizing the second-largest value, relevant for second-price auctions.

Strengths: The introduction of this paper is superb: it goes straight to the point and gives several (5) possible motivations for their problem, all convincing, and relevant to the NeurIPS community. They then clearly explain their contributions and precisely how they extend prior work, especially [KR18]. After skimming [KR18], I'm convinced that this submission presents a significant advance over that work. I enjoyed the discovery that there are algorithms that outperform the known lower-bound for score-based algorithms (the PTAS). I also liked the clever hardness proof.

Weaknesses: The experiments don't show a clean separation between different algorithms, and for practical purposes the choice of the best quantile is perhaps not quite settled.

Correctness: I believe so, but did not check everything carefully.

Clarity: The results are clearly explained, and intuitions for proof strategies are convincing. The writing is great.

Relation to Prior Work: Yes.

Reproducibility: Yes

Additional Feedback: line 68-69: "scored based" line 70: "This results" Footnote 1: "is MHR is" Theorem 3: Do you have an example showing that the KR18 scoring rule based on 1/k doesn't work for smax (in the sense of being constant-factor)? Theorem 3: Mention in the vicinity the KR18 result, and their factor 16. ######### ADDED ########## Apologies for my brief review, but I did not have much to complain about. Please make sure to invest some additional work into your experimental section. Your review contains the footnote "We did not try to optimize the constant factors." -- this could well become in a footnote in the real paper too.


Review 2

Summary and Contributions: The authors study the problem of selecting k out of n random variables to maximize either the expected highest value or the expected second highest value, motivated by welfare and revenue maximization problems in auctions with a limit on the number of bidders (among other applications). The authors show that when the distributions are discrete and provided explicitly, the objective of maximizing the expected highest value admits a PTAS, whereas the objective of maximizing the expected second highest value does not admit any constant factor approximation assuming either of Planted Clique or Exponential Time hypothesis. However, if the distributions are continuous and satisfy MHR, then they give a constant factor approximation for the latter objective. They also provide interesting simulations.

Strengths: * The problem is extremely elegant and has wide ranging applications. * The results are quite significant, non-trivial, and advance the state-of-the-art quite a bit. * I expect the results to be interesting to a wide cut of NeurIPS audience.

Weaknesses: * The only weakness of the paper, in my opinion, is the simulations. This paper did not really need them, and while the theory is very well executed, I find the simulations a bit haphazard, especially with the real-world data. It was very difficult to understand what the authors are doing here. What are the random variables here? How are they fed to the algorithms? If you're using neural networks, why do you call it a linear regression? (Even if you use a squared loss, the classifier learned by a neural network is not linear.) * Also, the authors exclude their PTAS, citing that it is not very practical, but do not mention how much time it takes to run it. If you include an experiments section, a reader would be very interested to know that. However, I believe that the authors can easily rewrite the portion about experiments with real-world data to clarify such details, and add what they found about the PTAS.

Correctness: The proofs are in the appendix, which I did not check, but based on the description provided in the paper, I find it believable that the results hold.

Clarity: Except for some parts of Section 6, the paper is written very clearly. The theoretical results are organized well.

Relation to Prior Work: While the authors did not include an explicit Related Work section, they did a decent job mentioning previous literature on this problem. The authors may want to check which other fields this problem has been investigated in under other names. For example, this problem has been studied in voting theory (see the paper below). There, it is assumed that instead of being given the distributions directly, there are a number of voters, and each voter observes a sample from each random variable and provides a ranking of the random variables. The goal is still to select k out of n variables to maximize the expected highest quality. This paper also seems to provide other motivations, which the authors may be able to use for their work too. "A Maximum Likelihood Approach For Selecting Sets of Alternatives", Procaccia et al., UAI 2012.

Reproducibility: Yes

Additional Feedback: * I think your title is quite narrow, given how widely applicable your problem is (perhaps your initial motivation to study this was auction theory?). How about a more representative title such as "Approximation Algorithms for Selecting a Subset of Random Variables" or something like that. I just think that more people will read your paper that way. * Do you know what is the best c such that using c/\sqrt{k} quantile gives the optimal approximation factor? * Do you know if the objective of maximizing the highest value is NP-hard? It would be good to add a future work section. You could remove some part of Section 6 for this, if you want. But this is a very subjective choice, so I leave it up to you. * Line 92: Missing S in i \in S * There are a number of things I would recommend to improve the math presentation. - Using \operatorname{OPT}, \operatorname{smax}, etc. - Using \frac and \nicefrac instead of inline fractions. - Using \cdot where appropriate, e.g., in the numerator of the equation in Line 140, or in Line 219. - Use \Pr, \log instead of Pr and log. - Using (Lemma 1) explanation right after a \ge sign seems to make the equation block look ugly. I would recommend writing afterwards in text or at the right hand with \because sign. * Line 152: So the running time is polylog(k). Earlier, you mentioned "nearly-linear". Is that nearly linear in the total description length of the RVs? It would be nice to add a comment here. * Section titles should be in title case. * Line 216: "the algorithm of [CD15]" -> the algorithm of Cai and Daskalakis [CD15] * Line 262: "bounded" -> finite-sized? * How many simulations is Figure 3 averaged over?


Review 3

Summary and Contributions: In this paper, the authors discuss how to select k out of n random variables so that the expected highest or second-highest value is maximized. For the highest value, they provide a PTAS algorithm; and for the second-highest value, they show hardness result for general case and give a constant approximation algorithm for the special case where the data has MHR. They also consider the case in ML where the true distribution is unknown and run experiments in this case.

Strengths: 1. For the highest value, the PTAS algorithm is not score-based, which circumvents the difficulty proved in KR18. 2. For the second-highest value, under MHR distribution, they provide a quantile-based algorithm which has constant approximation ratio.

Weaknesses: Most results in the paper require explicit descriptions of the random variables, which is not realistic in practice. This makes two difficulties in the application. 1. Performance: although the theoretical guarantee is good, the strong requirement of the explicit description of distribution makes the real performance in practice unclear. When the data is given, we need to learn either the simplified discrete distribution (section 3) or the expectation on the top q quantile (section 5). However, it is not clear if such criteria are estimated approximately from the data, how much will affect the performance. This will weaken the importance of the theoretical results in the paper. 2. Running time:Take the PTAS algorithm as the example. Though the running time of the PTAS algorithm is polynomial, it counts only the brute force process. It does not count the time to “simplify” the probability distribution, for example, to compute the probability of Hmax or REL(i), however, it might be complicated and time-consuming to compute it in practice. ADDED: For the explicit distribution, I will expect to see the analysis of inaccurate distribution, and hope that the theoretical result can still hold even if the distribution is learned from real massive data. Of course, this maybe deserve another (or several) NeurIPS paper in the future. Hope to see the follow-up work about it.

Correctness: Yes

Clarity: The writing of the paper can be improved. See the following detailed comments. 1. In line 122, what is H_max? 2. In line 127, is the support in the range [eps^2\tau,\tau]\cup {Hmax} \cup {0}? Otherwise, where is the probability of the outcome with value smaller than eps^2\tau? 3. I don’t understand step 4 and 5 in the pre-processing (line 132-140) without reading the supplementary. I think more explanation is needed for the meaning of decomposition and marginal contribution. 4. In section 4, it is better to briefly provide the definition of Densest k-subgraph problem. 5. In the reduction in section 4, what is p_e in line 169? 6. It seems lemma 1 does not depend on MHR. If so, it is better to clarify this point.

Relation to Prior Work: Yes

Reproducibility: Yes

Additional Feedback:


Review 4

Summary and Contributions: This paper considers selecting random variables to maximize the expected value of the maximum and second maximum of the selected variables. This paper provides several theoretical results for the problem. In particular, this paper proposes a PTAS for maximizing the largest expected value, proves the hardness of maximizing the second-largest expected value, and proposes a quantile-based algorithm under the monotone hazard rate assumption. The quantile-based algorithm was compared to several algorithms in the experiments.

Strengths: The considered problem is fundamental and would have a strong connection to this conference. As pointed out in Section 1, I think the general contribution of this paper contains certain differences from the previous work [KR18]; the proposed algorithms and hardness result seems to be novel in my knowledge.

Weaknesses: While there are important differences in the broad contributions, I feel that the significance of the proposed algorithms and experimental results is not so high. Specifically, I have several concerns about the contributions: - The proposed PTAS does not appear to be practical. In fact, this paper admits the weakness in lines 255 and 276. - While the derivation of the quantile-based algorithm would be a good contribution, the approximation factors are quite poor. In particular, I'm not sure how significant the factor 1000 of the second largest expected value is. - It's unclear how important the proposed algorithm is compared to the existing algorithm [KR18]. Of course, the differences are described in Section 1, the experimental results showed the KR algorithm was sometimes better than the proposed quantile-based algorithm. The paper did not provide an enough discussion of how to interpret the experimental results. - Related to the above concern, it seems that the experimental results do not strongly support the merits of the proposed method. From the results, it is unclear what the strengths of the proposed method are and thus what this paper wants to point out through the experiments.

Correctness: As far as I checked, I didn't find any serious mistakes in the theoretical part. In the experiments, one concern is the baseline of the Twitter data. Although the experimental setup is somewhat unclear, the proposed quantile-based algorithm and the KR algorithm use neural networks, whereas it seems that the baseline of the square loss uses a linear regression model. If so, it could be a rather unfair setup.

Clarity: The experimental protocol is quite unclear. There is no explanation of the neural network models and linear regression model used in the Twitter data experiment. The second experiment is suddenly conducted with a small amount of data, but there is no enough explanation of the motivation for conducting this experiment. Also, as pointed as Weaknesses, there is no sufficient explanation of how the results of the experiments should be interpreted.

Relation to Prior Work: It is good that Section 1 describes the broad differences from [KR18], but it should be explained in more detail. [KR18] contains a variety of results, and this paper refers to different results from [KR18]. Please clarify which results (theorems, lemmas, etc.) this paper refers to one by one. Also, it would be helpful if the detail of the KR algorithm in the experiments could be explained. Theorem 1 is proven in Section 3 and Appendix A, but this paper didn't cite any references. Is this algorithm entirely the original idea of this paper? If there are existing studies, I think it would be better to cite them.

Reproducibility: No

Additional Feedback: - In line 7, 'allows' -> 'allow'. - In line 42, 'of expected' -> 'of the expected'. - In line 43, 'expected' -> 'the expected'. - In line 51, 'with highest' -> 'with the highest'. - In line 60, 'scores' -> 'score'? - In line 68, '(value,probability)' -> '(value, probability)'. - In line 70, 'shows' -> 'show'. - In line 92, 'smax_{i \in}' -> 'smax_{i \in S}'. - In line 97, 'can we' -> 'we can'? - In line 105, 'Statistics' -> 'statistics'. - In line 126, 'expected' -> 'the expected'. - In line 128, '\log(1/\epsilon)' -> '\log(1/\epsilon))' (the right bracket is missing). - In line 154, 'expected maximum' -> 'the expected maximum of random variables'? - In line 163, 'Densest' -> 'densest'. - In line 177, 'expected' -> 'the expected'. - In line 178, 'expected' -> 'the expected'. - In line 202, 'as inputs' -> 'inputs as'? - In line 209, '\beta_1 and \beta_2' is preferred to '\beta_1, \beta_2'. - In line 214, 'takes value' -> 'takes a value'. - In line 261, 'U' is not defined. Is it exactly a uniform distribution? - In line 263, 'V_max' is not defined. - In line 278, 'expected' -> 'the expected'. - In line 305, 'use as features (distinct) words' seems to be not correct. - In line 308, 'least square loss' -> 'the square loss'? - In line 332, 'probability' -> 'the probability' - In line 333, 'normal' is preferred 'Gaussian' to be consistent in this paper. ----- Comments after the author feedback ----- The author feedback allayed my concerns about the experimental setting. It seems that I have misunderstood the description of linear regression, and so I realized that the experimental setup is not unfair for the baseline. I also believe that the next version will provide additional information about the experimental setting. On the other hand, I feel the experimental part is still weak to clarify the practical merits of the proposed algorithm. In particular, it would be better to conduct experiments that make clear the difference between the proposed algorithm and the KR algorithm. I also suggest the authors update the experimental part to explain the purpose and discuss the results more in the next version.

[Author Response · NeurIPS 2020]

# Author Response: Submission #8113

We thank all reviewers for the constructive feedback. We will incorporate the valuable suggestions from all reviewers (including a title change, as suggested by **R2**). In addition, we briefly address some of the comments below.

## Experiments

All reviewers gave feedback or had questions about the experimental set up. We briefly recap the experiments and comment on the purpose and importance of each one, as well as address some reviewer specific comments.

**Purpose:** The experiments highlight avenues toward bridging the gaps between the following standard theory assumptions and practice: (i) **Asymptotic run-time analysis:** Our PTAS is fantastic in theory but not practical. In practice we show that QR, which is very simple, is also near-optimal (Fig. 1). (ii) **Constant-factor approximation:**[1] In practice, Greedy, QR, and KR all perform extremely well (in fact, they have comparable behavior despite the very different theoretical guarantees) and much better than the baseline of Expectation (Fig. 1). (iii) **Availability of explicit distribution:** Thanks **R3** for bring this up! (See also Lines 292-4.) In practice explicit distributions typically only arise if we fit a model (e.g. Gaussians) to data. A slightly more realistic assumption are historical samples from the same distribution. In our real data experiments, we go one step further: We consider the practical scenario where we observe only one value for each feature vector. Here, we have an *implicit distribution over our uncertainty*. We develop a novel approach that allows us to successfully apply analogs of KR and QR in this setting (Fig. 3). (iv) **Just maximize the objective:** In theory, improving the objective function is always a better outcome. In practice, in particular in the context of the **broader impact** of ML research, it is important to explore the bias introduced by different algorithms. In particular we hypothesized that some algorithms for our problems will be biased by data scarcity, a well-documented bias in practical ML. In our second synthetic experiment each distribution gets a random label: $l$ (less data) or $m$ (more data); based on this label we only see a less or more samples from this distribution. In this experiment we measure the percentage of each population ($l$ vs $m$) selected by each algorithm, as well as performance (expected largest/second largest value). We see that while the performance is almost identical, the choice of method and quantile (both for KR and QR) has major effects on the percentage of small sample candidates selected.

**R2** asks what the random variables are in the real data experiment. The random variables are our implicit uncertainty about the value corresponding to a feature vector. We estimate the quantiles (resp. expectation) of this implicit distribution using quantile (resp. squared loss) regression. For all methods we use neural nets (and hopefully this clears up a confusion of **R4** regarding linear regression being treated differently: it isn't), and as **R2** correctly comments, "linear regression" should instead be "neural net with squared loss"; we have corrected this. **R4** asks about neural net models; we will provide additional information, including the depth, loss function and platform used to train these neural nets, as well as our methodology for training and testing. **R2** asks about the number of simulations in Fig. 3. Each feature vector/tweet in the test data is used only once. An experiment samples 500 tweets and picks $k$ of them (we give figures for multiple values of $k$), and Fig. 3 is averaged over 8000 experiments (the 4000 number is a typo).

As **R1 and R4** point out, the first part of the synthetic experiments and the Twitter data experiments don't give a clean separation between the algorithms. However, the surprise here is that despite the poor approximation guarantees of QR in theory, in practice it does just as well as the theoretically superior (better approximation guarantee without the MHR assumption, at least for expected maximum) KR algorithm. Furthermore, in terms of simplicity, the QR algorithm has the advantage as it has a *strict* subset of the steps of the KR algorithm. Finally, even though [KR18] introduced this score function (the score equals to the expectation over the top $1/k$ quantile), adapting it to the real data as two consecutive training steps (first quantile loss and then squared loss) is a contribution of this paper. We hope this addresses the comments of **R3 and R4** about the applicability, significance and advantages of the proposed method.

## Additional Comments

**R2** asks about the optimal $c/\sqrt{k}$ quantile. We do not know about the optimal one, but one can improve the approximation factor for the expected maximum objective by picking the $1/k$ quantile, using a similar analysis. Maximizing the expected maximum is indeed NP-hard; we will add this result.

**R2 and R4** ask about the running of the PTAS. The running time is $O(n|V|polylog(k))$, where $|V|$ is the maximum size of the support of a random variable, so "near-linear". This does, of course, take into account all operations, including simplifications, calculating conditional expectations and so on. We will clarify this.

## Footnotes

[1]We did not try to optimize the constant factors.


[Meta-Review · NeurIPS 2020]

All the reviewers and myself are in agreement that this paper provides a strong theoretical contribution on a general problem of broad appeal. The only minor comment is that the authors should try to improve in the final version, the presentation of the experimental part of their work.